# MMPerspective: Do MLLMs Understand Perspective? A Comprehensive Benchmark for Perspective Perception, Reasoning, and Robustness

**Yolo Yunlong Tang**[1,*], **Pinxin Liu**[1,*], **Zhangyun Tan**[1,*] **Mingqian Feng**[1], **Rui Mao**[1],
**Chao Huang**[1], **Jing Bi**[1], **Yunzhong Xiao**[2], **Susan Liang**[1], **Hang Hua**[1],
**Ali Vosoughi**[1], **Luchuan Song**[1], **Zeliang Zhang**[1], **Chenliang Xu**[1]

[1]University of Rochester, [2]Carnegie Mellon University

{yunlong.tang, mingqian.feng, jing.bi, chenliang.xu}@rochester.edu,
{pliu23, rmao6, lsong11, zzh136}@ur.rochester.edu,
ztan12@u.rochester.edu, {chuang65, sliang22, hhua2}@cs.rochester.edu,
avosoughi@ece.rochester.edu, yunzhonx@andrew.cmu.edu

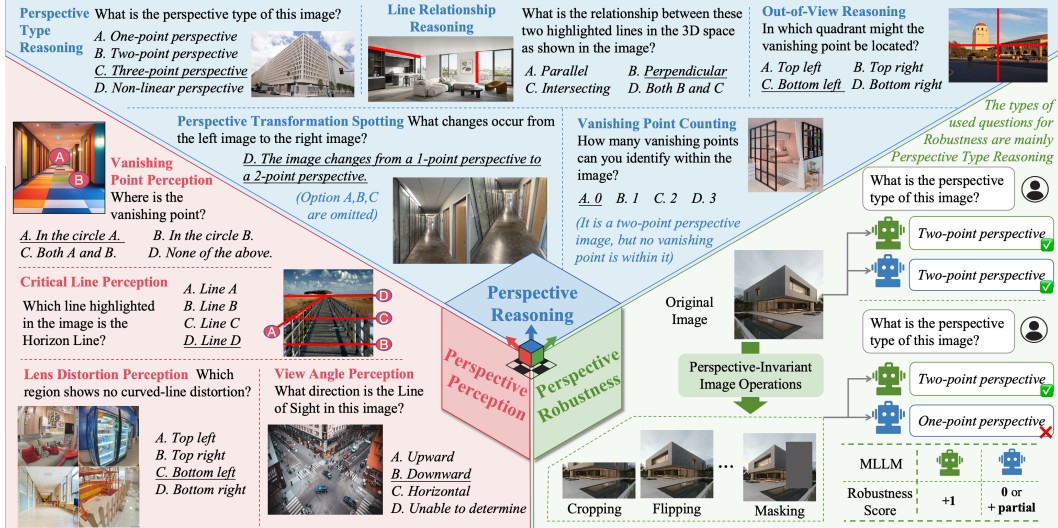

Figure 1: **MMPerspective benchmark overview.** We introduce 10 tasks spanning 3 complementary dimensions of perspective understanding: Perspective **Perception**, **Reasoning**, and **Robustness**.

## Abstract

Understanding perspective is fundamental to human visual perception, yet the extent to which multimodal large language models (MLLMs) internalize perspective geometry remains unclear. We introduce MMPerspective, the first benchmark specifically designed to systematically evaluate MLLMs' understanding of perspective through 10 carefully crafted tasks across three complementary dimensions: Perspective Perception, Reasoning, and Robustness. Our benchmark comprises 2,711 real-world and synthetic image instances with 5,083 question-answer pairs that probe key capabilities, such as vanishing point perception and counting, perspective type reasoning, line relationship understanding in 3D space, invariance to

*Equal contribution.

39th Conference on Neural Information Processing Systems (NeurIPS 2025) Track on Datasets and Benchmarks.

perspective-preserving transformations, etc. Through a comprehensive evaluation of 43 state-of-the-art MLLMs, we uncover significant limitations: while models demonstrate competence on surface-level perceptual tasks, they struggle with compositional reasoning and maintaining spatial consistency under perturbations. Our analysis further reveals intriguing patterns between model architecture, scale, and perspective capabilities, highlighting both robustness bottlenecks and the benefits of chain-of-thought prompting. MMPerspective establishes a valuable testbed for diagnosing and advancing spatial understanding in vision-language systems. Resources are available at `https://yunlong10.github.io/MMPerspective/`

# 1 Introduction

*Perspective is nothing more than a rational demonstration applied to the consideration of how objects in front of the eye transmit their image to it.*
— Leonardo da Vinci, *The Notebooks of Leonardo da Vinci* [Da Vinci, 2012]

From the chalked strings of Renaissance artists to the calibrated optics of modern cameras, perspective has long served as a cornerstone for representing three-dimensional reality on two-dimensional surfaces [Kemp et al., 1990, Neher, 2005]. Based on the geometry of the pinhole camera model, perspective projection enables humans to infer spatial structure, depth, and layout from flat images, a capability central to artistic creation, scientific visualization, and machine perception [Hartley, 2003, Hecht, 2012]. For instance, artists employ perspective to enhance realism, guide viewer attention, manipulate spatial illusion, and convey narrative depth [Robertson and Bertling, 2013, Panofsky, 2020]. In scientific visualization, perspective projections are used to render complex 3D structures, such as molecular surfaces and anatomical forms [Ware, 2019]. In computer vision, some methods based on the perspective principle have been developed to analyze, edit images, and fix distortions [Criminisi et al., 2002, Carroll et al., 2010, Carroll, 2013]. Therefore, perspective understanding plays a foundational role in visual cognition and spatial representation. However, current research [Bharadwaj et al., 2025, Coudert et al., 2022, Zhao et al., 2021] is still primarily focused on using perspective principles to implement various applications, with relatively little research on the ability of intelligent systems themselves to understand perspective. Although some studies have already attempted to enable models to locate vanishing points [Bharadwaj et al., 2025], detect key lines in space [Coudert et al., 2022, Zhao et al., 2021], etc., these models either rely on precise mathematical models or learn from specialized datasets, being hard to capture perspective-related semantics or apply their learned understanding of perspective to other more general tasks.

On the other hand, recent multimodal large language models (MLLMs) such as GPT-4o [Achiam et al., 2023] and Gemini [Reid et al., 2024] have demonstrated powerful human-like visual perception and reasoning capabilities through large-scale training, but their ability to understand perspective has not yet been tested. Given its foundational role in visual cognition and spatial representation, an important open question is: **Do MLLMs understand perspective?** These models have shown remarkable performance across a broad range of high-level vision-language tasks, including visual captioning [Wang et al., 2023] and visual question answering [Liu et al., 2024a, Achiam et al., 2023, Reid et al., 2024, Chen et al., 2024b, Wang et al., 2024b]. However, existing benchmarks rarely evaluate their capacity for geometric reasoning. In particular, it remains unclear whether MLLMs can identify vanishing points, understand the convergence of parallel lines, reason about spatial relationships induced by perspective, or maintain consistent spatial interpretations across different viewpoints. These are fundamental aspects of human visual understanding and have been systematically studied in both art history and computational vision [Robertson and Bertling, 2013], yet they are largely absent from current evaluation protocols [Yu et al., 2024, Liu et al., 2025, Li et al., 2024c, Hua et al., 2024, Wang et al., 2024d, Tang et al., 2024, 2025b,a] for MLLMs.

To bridge this gap, we introduce **MMPerspective**, the first benchmark specifically designed to evaluate perspective understanding in MLLMs. As shown in Figure 1, our benchmark comprises 10 tasks divided across three dimensions: **Perspective Perception**, **Perspective Reasoning**, and **Perspective Robustness**. Perception tasks probe the ability to identify geometric cues such as vanishing points and critical lines. Reasoning tasks examine models' ability to interpret 3D structure, assess scene composition, and predict off-canvas geometry. Robustness task evaluates spatial consistency under appearance-preserving transformations, such as flipping and cropping.

Our benchmark comprises **2,711** image instances and **5,083** question-answer pairs, each framed as a multiple-choice question grounded in real-world imagery rich with architectural, urban, and indoor perspective cues, such as vanishing lines, orthogonal edges, and depth gradients. Tasks are organized to increase in difficulty across perceptual, reasoning, and robustness dimensions, requiring progressively deeper spatial abstraction. We evaluate 43 state-of-the-art MLLMs, ranging from lightweight open-source models to proprietary systems like GPT-4o and Gemini. While many models perform competitively on surface-level perception tasks, they exhibit clear performance drops on reasoning and robustness tasks. For instance, models often fail to maintain consistent predictions under simple geometric-preserving edits, such as horizontal flipping or partial occlusion of key cues, revealing their limited internalization of spatial priors and geometric constraints.

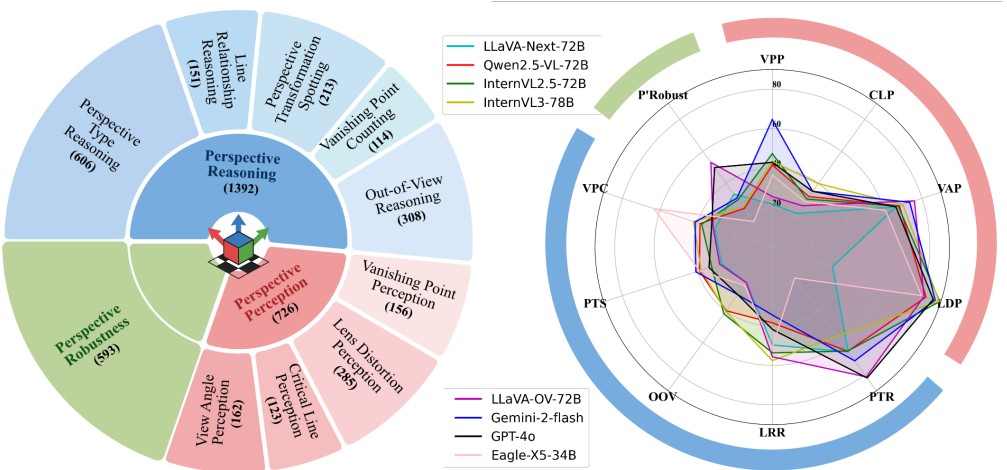

Figure 2: **Left**: MMPerspective benchmark consists of 2,711 instances and 5,083 QA pairs, hierarchically organized into 3 core categories (Perspective **Perception**, **Reasoning**, and **Robustness**). **Right**: The accuracy of 8 representative MLLMs on 10 tasks of MMPerspective across the 3 categories.

In short, our contributions are three-fold:

- We introduce **MMPerspective**, the first dedicated benchmark for evaluating perspective understanding in MLLMs, spanning 10 tasks across three dimensions, consisting of 2,711 instances and 5,083 QA pairs.
- We conduct a comprehensive evaluation of 43 representative MLLMs and reveal key limitations in perspective perception, reasoning, and robustness.
- We offer new insights into current model bottlenecks and provide guidance toward building geometry-aware, spatially grounded multimodal systems.

## 2 MMPerspective

### 2.1 Preliminary

Understanding the key elements of perspective geometry is essential for interpreting spatial relationships in 2D images. In this section, we introduce foundational terms used, following classical principles of linear perspective as described in drawing literature [Robertson and Bertling, 2013]. As shown in the Figure 3, the **Ground Plane (GP)** is the surface upon which objects rest and from which vertical height is measured. The **Station Point (SP)** represents the viewer's position in space, typically aligned with the eye or camera origin. The

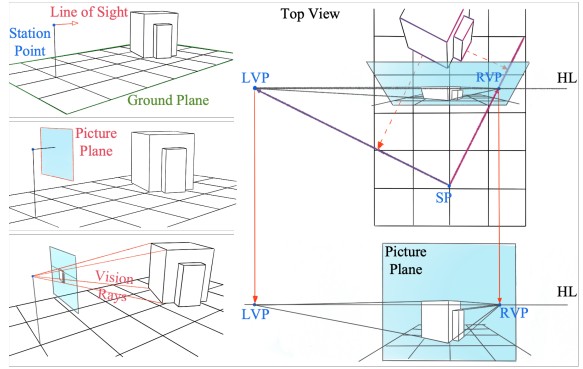

Figure 3: Perspective illustration with terminology. The figure is adapted from [Robertson and Bertling, 2013].

**Line of Sight (LS)** defines the direction in which the observer is looking; when this is parallel to GP, vertical lines in the scene remain vertical in the image, as seen in one- or two-point perspectives. Tilting the LS results in three-point perspective, where verticals also converge. The **Picture Plane (PP)** refers to an imaginary plane perpendicular to the LS where the visual projection occurs. It is often conceptualized as a transparent sheet placed between the observer and the scene, capturing the intersections of visual rays from the Station Point to the object. The **Vision Rays (VRs)** are the lines extending from the eye through each point on the object to the PP. The **Horizon Line (HL)** corresponds to the viewer's eye level and is the projection of the GP onto the PP. A **Vanishing Point (VP)** is the point at which a set of parallel lines appears to converge. In 1-point perspective, a single set of lines converges to one VP. In 2-point perspective, two sets of lines converge to separate VPs on the HL. In 3-point perspective, an additional VP is used for vertical convergence, located either above or below the HL, depending on whether the observer is looking up or down.

## 2.2 Taxonomy

The MMPerspective benchmark is designed to evaluate perspective understanding in MLLMs across three complementary and hierarchically structured dimensions: **Perspective Perception**, **Perspective Reasoning**, and **Perspective Robustness**. These dimensions reflect a progression from low-level visual recognition to high-level spatial inference and consistency under image transformations.

**Perspective Perception (P'Percep)** focuses on a model's ability to detect and interpret explicit perspective-related cues directly visible in the image. It includes the following tasks: **Vanishing Point Perception (VPP)** evaluates whether a model can correctly locate a VP or determine its presence within a given region. **Critical Line Perception (CLP)** assesses the identification of the HL from a set of candidate lines, based on perspective convergence. **Lens Distortion Perception (LDP)** requires the model to distinguish regions in the image that are free from curved-line distortion. **View Angle Perception (VAP)** asks the model to infer the LS direction (e.g., upward, downward, or horizontal) using visible spatial cues. All tasks in this category are grounded in localized, directly observable visual evidence and require minimal reasoning beyond geometric feature detection.

**Perspective Reasoning (P'Reason)** tests whether the model can integrate multiple spatial cues and apply geometric reasoning to infer high-level relationships in the 3D structure of the scene. The tasks include: **Perspective Type Reasoning (PTR)**, which involves classifying the underlying perspective structure of the image (e.g., 1-point, 2-point, 3-point, or non-linear). **Line Relationship Reasoning (LRR)**, which asks the model to determine whether two lines in the 3D space are parallel, perpendicular, or intersecting. **Perspective Transformation Spotting (PTS)**, which requires detecting changes in perspective type across paired images. **Vanishing Point Counting (VPC)**, which involves estimating the number of identifiable VPs present in the scene. **Out-of-View Reasoning (OVR)**, which challenges the model to infer the quadrant in which a VP lies when it is not explicitly shown in the image. These tasks demand a combination of compositional reasoning, global geometric understanding, and spatial abstraction beyond direct visual perception.

**Perspective Robustness (P'Robust)** assesses the model's ability to produce consistent and geometry-aware predictions under controlled, appearance-preserving transformations of the input image. Each original image-question pair is augmented with perturbed versions through perspective-invariant operations such as cropping, flipping, and masking. While these transformations do not alter the scene's underlying geometry, they may obscure or de-emphasize key visual cues. A model is considered robust if it provides the same, correct answer across all such transformed variants. This consistency serves as a direct measure of its geometric grounding, separating genuine perspective understanding from brittle reliance on surface-level visual patterns.

## 2.3 Data Curation

**Data Collection.** To support the construction of these tasks, we curated a diverse set of perspective-rich images from multiple sources. Images are sourced from four streams. **First**, we collect unlabeled examples from the web, primarily architectural and indoor scenes with strong perspective cues. **Second**, we shoot real-world perspective images in life scenarios with both linear perspectives and curvilinear perspectives (fish-eye perspectives). For one scene, we shoot multiple images with different views to form perspective image pairs. **Third**, we incorporate data from the open-source RPVP datasets [Bharadwaj et al., 2025]. In this dataset, perspective cues come from the recurrence

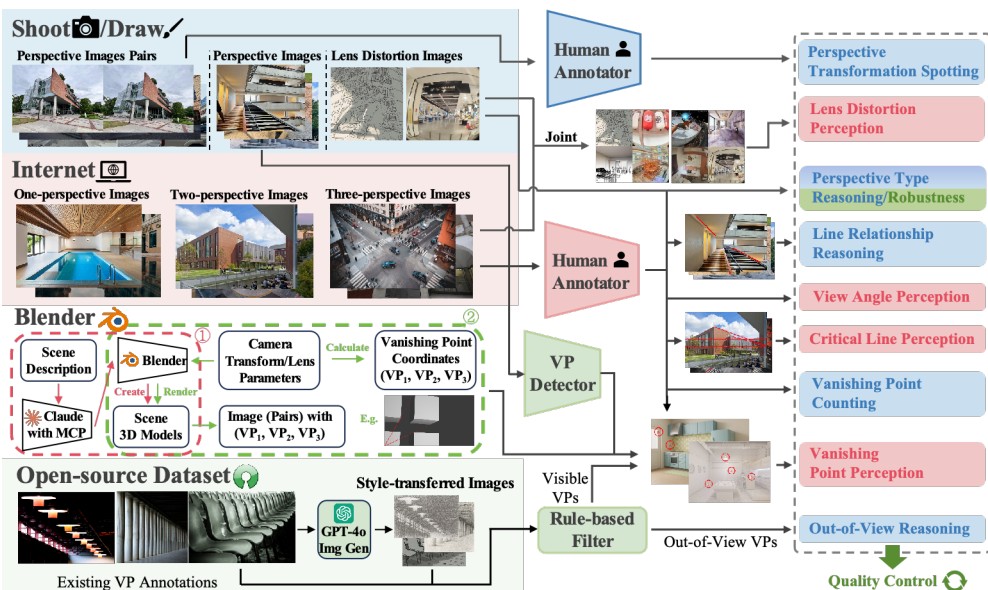

Figure 4: Data Curation Pipeline for MMPerspective.

pattern rather than lines at object edges. **Fourth**, we utilize Blender to create images with ground-truth VP coordinates. Specifically, we first employ Claude 3.7 Sonnet to create 3D models based on scene descriptions, empowered by Blender-MCP. For each scene, we render multiple images with different camera transform and lens parameters. From these parameters, we calculate the ground-truth VP coordinates for each image. We provide more details of this approach in Appendix.

**Annotation.** We annotate each image with task-specific metadata using a hybrid pipeline. For **PTS**, we manually annotate the perspective changes in the image pairs that we shoot. For **LDP**, we combine fish-eye perspective images and regular linear perspective images randomly and record the corresponding option. For **PTR**, **LRR**, **VAP**, **CLP**, and **VPC**, we use images collected from the web and manually annotate the right answers for the questions and hints on the images. For **VPP**, we use both images from the web and Blender. The VP annotations of the former are manually created, while the latter are born with ground-truth VP coordinates. For **OVR**, we use the annotation from the RPVP datasets [Bharadwaj et al., 2025].

**Quality Control.** Quality assurance is carried out via a multi-stage review process. All automatically generated annotations are verified manually. For subjective tasks involving spatial reasoning, at least two annotators independently label each sample, with disagreements resolved through discussion and consensus. We exclude any examples where ambiguity could not be resolved, and the final benchmark comprises only unambiguous, perspective-defining scenes. We also manually check and filter all unsafe images we collect.

## 2.4 Evaluation Metrics

For all tasks in **P'Percep** and **P'Reason** of MMPerspective, we use accuracy as the main evaluation metric, where each question has one correct answer. For **P'Robust**, we evaluate consistency under image perturbations and report two complementary metrics:

**Binary P'Robust Score.** Let $\mathcal{S}$ be the set of robustness seed items. For each seed $(I_s, q, a^*) \in \mathcal{S}$ we consider the set of images $V_s = \{I_s\} \cup \{I_1, \ldots, I_{n_s}\}$ which includes the original image and all its perturbed variants. Binary robustness requires perfect consistency across all images in $V_s$:

$$\text{Binary-Robust}_{\mathcal{M}} = \frac{1}{|\mathcal{S}|} \sum_{(I_s, q, a^*) \in \mathcal{S}} \mathbb{1}\left[\bigwedge_{I \in V_s} \mathcal{M}(I, q) = a^*\right]. \tag{1}$$

**Graded P'Robust Score.** To capture partial consistency, we additionally compute a graded score that averages the fraction of correctly answered images within each set $V_s$:

$$\text{Graded-Robust}_{\mathcal{M}} = \frac{1}{|\mathcal{S}|} \sum_{(I_s, q, a^*) \in \mathcal{S}} \left( \frac{1}{|V_s|} \sum_{I \in V_s} \mathbb{1}[\mathcal{M}(I, q) = a^*] \right). \tag{2}$$

For example, if a model answers $4$ out of $5$ images in $V_s$ correctly, its per-set graded score is $0.8$, while its binary score for that set would be $0$.

## 3 Experiments

### 3.1 Experiment Setup

We select 20 representative models, including both open-source and proprietary models, covering a broad spectrum of model scales and architecture types. These include GPT-4o [OpenAI et al., 2024], Gemini-2 [DeepMind, 2025], LLaVA-OV [Li et al., 2024b], LLaVA-Next [Liu et al., 2024b], InternVL2 [Chen et al., 2024b], InternVL2.5 [Chen et al., 2024b], InternVL3 [Zhu et al., 2025], Qwen2-VL [Wang et al., 2024c], Qwen2.5-VL [Bai et al., 2025], and Eagle-X [Shi et al., 2024]. To ensure fairness and eliminate potential positional bias, we have already randomly shuffled the answer choices for all questions during the dataset creation process. To ensure consistency, all open-source models under 14B are evaluated using a single NVIDIA A6000 48GB GPU. Models larger than 14B and up to 70B are evaluated using a single NVIDIA H100 80GB GPU. Larger models (>70B) are run on multiple NVIDIA A100 80G GPUs (at least 4). Proprietary models are executed via APIs. Each model is evaluated under the same test conditions, with identical multiple-choice question formats across all tasks. To ensure deterministic and fully reproducible results for all our experiments, we employed a greedy decoding strategy for all open-source models. For proprietary models accessed via API, we also used their deterministic decoding modes where available. This approach eliminates randomness from the decoding process, ensuring that a model's output for any given sample is consistent across multiple runs.

### 3.2 Main Results

Table 1 presents the performances of various MLLMs on our MMPERSPECTIVE benchmark. In general, larger models tend to perform better, with GPT-4o and Gemini-2-flash achieving the highest overall accuracy (57.7% and 57.6%, respectively).

**Perspective Perception.** For **VPP**, Gemini-2-flash (CoT) achieves the highest accuracy (69.8%), while many smaller models struggle with this fundamental task. In **CLP**, all models perform poorly, with even GPT-4o (CoT) only reaching 46.3%, indicating a general limitation in detecting HLs. Most larger models exceed 60% on **VAP**, with InternVL3-14B leading at 73.5%. For **LDP**, InternVL3-38B demonstrates the strongest performance (90.9%), surpassing even proprietary models.

**Perspective Reasoning.** In **PTR**, GPT-4o achieves the highest score (82.0%), with LLaVA-OV-72B close behind (81.4%). **LRR** shows less correlation with model size, with InternVL3-78B leading at 57.6%. For **OVR**, InternVL3-38B significantly outperforms all others (56.8%), suggesting unique architectural advantages. In **VPC**, the Eagle-X4 family demonstrates superior performance (68.4% for 8B), indicating specialized capabilities for identifying multiple VPs.

**Perspective Robustness.** **P'Robust** scores reveal surprising patterns, with Eagle-X4-8B achieving great performance (55.3%) despite modest size. LLaVA-OV-72B (53.1%) and Eagle-X4-13B (53.8%) also present strong robustness. Notably, many large models with high accuracy perform poorly on robustness, with InternVL3-38B showing excellent perception (67.2%) but poor robustness (9.1%).

### 3.3 Further Findings

> **Finding 1.** Our analysis reveals that perspective understanding scales strongly with total model size but only weakly with vision encoder size, with robustness showing particularly limited correlation to encoder scaling.

Our analysis of model scaling reveals important insights into how different architectural components influence perspective understanding capabilities in MLLMs. In Fig. 5, there is a clear progression of

Table 1: **Performance of MLLMs on MMPerspective.** Models are grouped by size and ranked by overall accuracy. Best scores in each group are bolded.

| Model | Perspective Perception | | | | Perspective Reasoning | | | | | P'Percep & P'Reason | | | Robustness | |
|---|---|---|---|---|---|---|---|---|---|---|---|---|---|---|
| | VPP | CLP | VAP | LDP | PTR | LRR | OVR | PTS | VPC | P Acc | R Acc | Overall | Graded | Binary |
| *MLLMs: < 7B* | | | | | | | | | | | | | | |
| InternVL2.5-2B | **47.4** | 22.8 | 13.0 | **65.3** | **62.2** | 31.8 | 16.6 | 30.0 | **50.0** | 37.1 | **38.1** | **37.7** | **59.1** | **46.5** |
| Qwen2.5-VL-3B | 27.6 | 22.8 | 56.8 | 55.1 | 32.3 | 32.5 | 15.9 | **39.4** | 44.7 | 40.6 | 33.0 | 36.3 | 22.2 | 6.4 |
| InternVL2.5-4B | 32.1 | 26.0 | **59.3** | 64.2 | 28.2 | 30.5 | 10.7 | 37.1 | 36.8 | **45.4** | 28.7 | 36.1 | 25.0 | 20.6 |
| InternVL3-2B | 22.4 | **28.5** | 50.0 | 44.6 | 43.1 | 31.1 | **34.4** | 25.4 | 43.0 | 36.4 | 35.4 | 35.8 | 39.0 | 23.9 |
| InternVL2-4B | 26.9 | 12.2 | 54.3 | 60.4 | 18.0 | **40.4** | 18.8 | 24.4 | 45.6 | 38.4 | 29.4 | 33.4 | 14.5 | 7.9 |
| Qwen2-VL-2B | 12.2 | 19.5 | 49.4 | 35.8 | 23.3 | 24.5 | 28.9 | 32.9 | 47.4 | 29.2 | 31.4 | 30.4 | 18.0 | 4.7 |
| InternVL3-1B | 19.9 | 13.0 | 53.7 | 20.7 | 16.3 | 8.6 | 23.7 | 21.6 | 47.4 | 26.8 | 23.5 | 25.0 | 16.1 | 13.8 |
| InternVL2-1B | 20.5 | 20.3 | 15.4 | 24.2 | 24.1 | 11.3 | 24.0 | 22.1 | 44.7 | 20.1 | 25.2 | 23.0 | 18.2 | 6.7 |
| LLaVA-OV-1B | 13.5 | 14.6 | 35.8 | 24.2 | 15.2 | 19.2 | 19.5 | 22.1 | 40.4 | 22.0 | 23.3 | 22.7 | 13.0 | 7.8 |
| InternVL2-2B | 26.9 | 26.0 | 3.1 | 36.8 | 18.8 | 12.6 | 23.1 | 21.1 | 34.2 | 23.2 | 22.0 | 22.5 | 19.3 | 12.3 |
| InternVL2.5-1B | 14.7 | 23.6 | 0.6 | 33.0 | 20.1 | 11.3 | 13.3 | 34.7 | 45.6 | 18.0 | 25.0 | 21.9 | 19.0 | 18.2 |
| *MLLMs: 7B - 9B* | | | | | | | | | | | | | | |
| InternVL2.5-8B | 38.5 | 17.9 | 53.1 | 75.4 | 40.8 | 48.3 | 34.7 | 24.9 | 67.5 | 46.2 | 43.3 | **44.6** | 38.7 | 22.3 |
| Qwen2.5-VL-7B | 35.3 | 29.3 | **70.4** | 73.7 | 42.4 | 44.4 | 32.1 | 28.6 | 44.7 | 52.1 | 38.5 | 44.5 | 33.2 | 15.3 |
| Qwen2-VL-7B | 34.6 | 25.2 | 63.0 | 64.2 | 57.1 | 49.0 | 27.3 | 31.0 | 46.5 | 46.7 | 42.2 | 44.2 | 46.9 | 25.5 |
| InternVL3-9B | 37.2 | **33.3** | 63.0 | 77.5 | 30.7 | **53.0** | 27.9 | 23.9 | 43.9 | 52.8 | 35.9 | 43.4 | 19.2 | 7.3 |
| InternVL3-8B | **42.3** | 27.6 | 67.9 | **81.8** | 38.1 | 46.4 | 20.8 | 23.9 | 32.5 | **54.9** | 32.3 | 42.4 | 29.1 | 15.9 |
| LLaVA-OV-7B | 34.0 | **33.3** | 51.2 | 57.9 | 44.9 | **53.0** | 19.8 | **35.2** | 49.1 | 44.1 | 40.4 | 42.0 | 36.1 | 15.9 |
| Eagle-X4-8B | 39.1 | 17.1 | 46.9 | 47.7 | **65.3** | 37.1 | 18.2 | 32.9 | **68.4** | 37.7 | **44.4** | 41.4 | **60.7** | **55.3** |
| InternVL2-8B | 33.3 | 19.5 | 59.3 | 73.3 | 27.1 | 36.4 | **42.5** | 22.1 | 48.2 | 46.4 | 35.3 | 40.2 | 19.9 | 7.9 |
| LLaVA-Next-m-7B | 35.9 | 21.1 | 35.2 | 50.5 | 17.7 | 37.7 | 15.6 | 27.2 | 46.5 | 35.7 | 28.9 | 31.9 | 17.9 | 16.4 |
| Eagle-X5-7B | 25.0 | 26.0 | 24.7 | 34.7 | 22.1 | 46.4 | 15.6 | 20.7 | 42.1 | 27.6 | 29.4 | 28.6 | 18.4 | 15.9 |
| LLaVA-Next-v-7B | 16.7 | 20.3 | 40.7 | 39.6 | 16.3 | 44.4 | 19.8 | 16.4 | 7.0 | 29.3 | 20.8 | 24.6 | 16.7 | 16.4 |
| *MLLMs: 10B - 30B* | | | | | | | | | | | | | | |
| InternVL2.5-26B | 41.7 | **35.0** | 55.6 | **81.8** | 65.5 | **46.4** | 43.5 | **34.3** | 46.5 | **53.5** | **47.2** | **50.0** | 52.9 | 33.7 |
| InternVL3-14B | 39.1 | 26.0 | **73.5** | 73.3 | 36.5 | 34.4 | **54.5** | 28.2 | 54.4 | 53.0 | 41.6 | 46.7 | 27.3 | 13.5 |
| InternVL2-26B | 28.2 | **35.0** | 61.1 | 74.0 | 50.7 | 41.7 | 28.9 | 28.6 | 43.0 | 49.6 | 38.6 | 43.5 | 44.1 | 26.5 |
| Eagle-X4-13B | **42.3** | 26.8 | 41.4 | 44.6 | 65.8 | 20.5 | 28.2 | 31.0 | **57.9** | 38.8 | 40.7 | 39.8 | **60.7** | **53.8** |
| LLaVA-Next-13B | 7.7 | 17.1 | 54.3 | 34.7 | **66.7** | 24.5 | 13.0 | 26.8 | 43.9 | 28.5 | 35.0 | 32.1 | 59.7 | 51.1 |
| *MLLMs: 30B - 70B* | | | | | | | | | | | | | | |
| InternVL2.5-38B | **46.8** | 36.6 | 67.9 | 89.5 | 58.4 | 51.7 | 38.3 | **44.1** | 44.7 | 60.2 | **47.5** | **53.1** | 41.6 | 19.1 |
| InternVL3-38B | 45.5 | 35.0 | **71.0** | **90.9** | 37.3 | 43.0 | **56.8** | 37.6 | 43.0 | **60.6** | 43.5 | 51.1 | 23.9 | 9.1 |
| Qwen2.5-VL-32B | 35.9 | 22.8 | 68.5 | 73.7 | **62.0** | 37.7 | 33.8 | 35.2 | 45.6 | 50.2 | 42.9 | 46.1 | **48.8** | **25.5** |
| Eagle-X5-34B | 36.5 | 28.5 | 60.5 | 79.6 | 19.5 | 51.0 | 24.0 | 39.0 | **63.2** | 51.3 | 39.3 | 44.6 | 18.7 | 16.0 |
| InternVL2-40B | 26.3 | 22.0 | 66.0 | 76.1 | 43.2 | **55.0** | 27.3 | 25.8 | 47.4 | 47.6 | 39.7 | 43.2 | 29.5 | 12.6 |
| *MLLMs: > 70B* | | | | | | | | | | | | | | |
| InternVL3-78B | 43.6 | **39.8** | 69.8 | 89.1 | 55.9 | **57.6** | 40.3 | 38.0 | **42.1** | **60.6** | 46.8 | **52.9** | 43.6 | 25.5 |
| InternVL2.5-72B | **47.4** | 30.1 | 67.3 | **89.5** | 65.2 | 53.6 | **41.9** | 32.4 | 37.7 | 58.6 | 46.2 | 51.7 | 56.3 | 29.7 |
| Qwen2.5-VL-72B | 41.7 | 31.7 | 67.9 | 82.1 | 65.3 | 38.4 | 39.9 | **39.0** | 38.6 | 55.8 | 44.3 | 49.4 | 49.9 | 24.3 |
| Qwen2-VL-72B | 34.6 | 18.7 | 70.4 | 82.5 | 68.8 | 52.3 | 38.6 | 35.2 | **42.1** | 51.5 | **47.4** | 49.2 | 51.3 | 25.0 |
| LLaVA-OV-72B | 25.6 | 26.0 | **75.9** | 81.1 | **81.4** | 55.6 | 22.4 | 28.2 | 31.6 | 52.2 | 43.0 | 47.5 | **71.8** | **53.1** |
| LLaVA-Next-72B | 21.8 | 21.1 | 66.0 | 32.3 | 65.7 | 49.7 | 22.4 | 27.2 | 30.7 | 35.3 | 39.1 | 37.4 | 55.6 | 33.2 |
| InternVL2-72B | 26.9 | 18.7 | 57.4 | 56.8 | 56.1 | 47.0 | 24.7 | 24.4 | 7.9 | 40.0 | 32.0 | 35.6 | 43.9 | 22.9 |
| *MLLMs: Proprietary* | | | | | | | | | | | | | | |
| Gemini-2-flash (CoT) | **69.2** | 49.6 | 72.8 | 87.4 | 78.7 | 32.5 | **40.9** | 39.9 | 43.9 | **69.8** | **47.2** | **57.2** | 50.5 | 24.8 |
| GPT-4o (CoT) | 45.5 | 46.3 | 70.4 | **88.8** | 81.4 | **47.0** | 34.4 | 37.6 | 34.2 | 62.7 | 46.9 | 54.0 | 69.4 | **49.9** |
| Gemini-2-flash | 64.7 | 35.0 | **73.5** | 87.0 | 71.3 | 34.4 | 29.9 | **40.8** | 41.2 | 65.0 | 43.5 | 53.1 | 56.8 | 30.7 |
| GPT-4o | 42.9 | 35.0 | 66.0 | 86.0 | **82.0** | 41.7 | 29.9 | 33.8 | 32.5 | 57.5 | 44.0 | 50.0 | **71.9** | **49.9** |
| Gemini-1.5-flash (CoT) | 30.1 | 28.5 | 66.7 | 79.3 | 51.0 | 39.7 | 20.1 | 31.5 | 35.1 | 51.1 | 35.5 | 42.4 | 37.8 | 11.6 |
| GPT-4o-mini | 35.3 | 24.4 | 43.2 | 71.6 | 43.1 | 29.8 | 14.6 | 31.0 | **45.6** | 43.6 | 32.8 | 37.6 | 28.7 | 10.8 |
| Gemini-1.5-flash | 26.9 | 25.2 | 59.3 | 70.5 | 26.4 | 27.8 | 18.2 | 26.8 | 22.8 | 45.5 | 24.4 | 33.8 | 20.6 | 10.6 |

performance within model families as model size increases, with deeper blue coloration indicating higher accuracy and robustness for larger variants. The scatter plots in Fig. 6 quantify these relationships more precisely, demonstrating a strong positive correlation between model size and perspective understanding accuracy ($r = 0.81$), while robustness shows a weaker correlation ($r = 0.34$).

This disparity suggests that while general perspective understanding capabilities scale reliably with language model size, robustness to perspective-preserving transformations follows a different pattern. For instance, models like Eagle-X4 achieve high perspective robustness even at moderate sizes (8B and 13B), suggesting their architecture may have inherent advantages for maintaining consistent geometric interpretations across image variations.

When examining vision encoder scaling specifically (Fig. 6c-d), we observe a moderate correlation with overall perspective accuracy ($r = 0.51$) but a notably weak correlation with perspective robustness ($r = 0.15$). This suggests that vision encoders play a more limited role in ensuring consistent geometric interpretations across transformations than in enabling basic perspective understanding.

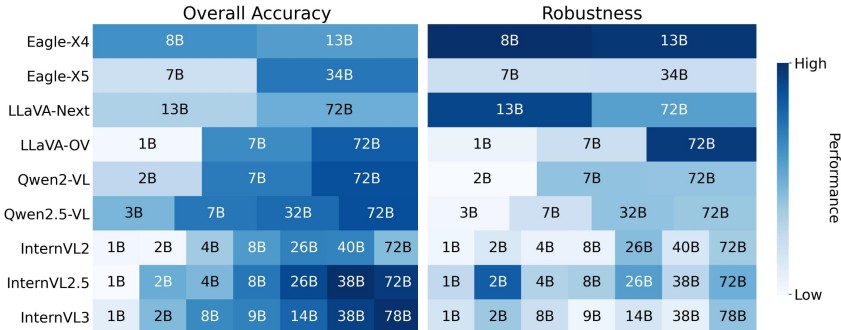

Figure 5: Heatmaps illustrating the relationship between model size and performance, measured by P&R Overall Accuracy and Robustness. Darker colors indicate higher performance. Each line represents a model family, with sizes increasing from left to right.

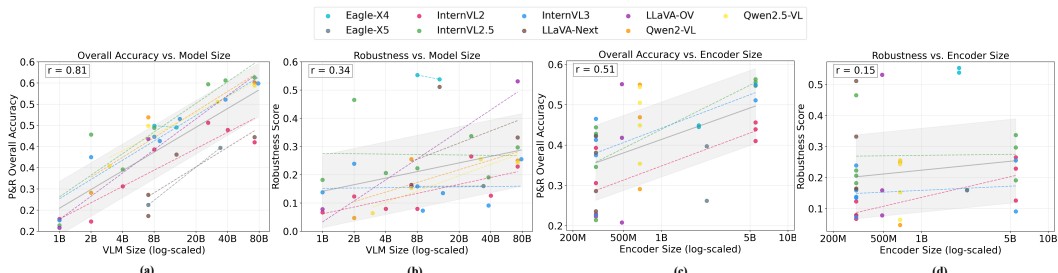

Figure 6: Correlation analysis between performance and size across MLLM families: (a) Overall accuracy vs. model size ($r = 0.81$), (b) Robustness vs. model size ($r = 0.34$), (c) Overall accuracy vs. encoder size ($r = 0.51$), (d) Robustness vs. encoder size ($r = 0.15$). Total model scaling strongly impacts perspective understanding, while vision encoder size has a limited influence on robustness.

The data indicates that while increasing vision encoder capacity may help models better recognize perspective features initially, it does not necessarily translate to more stable geometric interpretations when those features are partially obscured or repositioned.

The limited range of encoder sizes currently employed across model families (mostly 300-500M parameters) makes it difficult to draw definitive conclusions about vision encoder scaling laws for perspective understanding. This represents a gap in our understanding of how to optimally design MLLMs for spatial reasoning tasks that require both accurate perspective perception and consistent geometric interpretations under varying conditions.

> **Finding 2.** Chain-of-thought (CoT) prompting modestly improves model performance and robustness on perspective-related tasks by encouraging stepwise deduction.

As shown in Table 2, CoT prompting leads to consistent performance gains across nearly all perspective-related tasks. All three evaluated models, GPT-4o, Gemini-1.5-flash, and Gemini-2-flash, experience improvements in both perception and reasoning sub-tasks when CoT is applied. Notably, no single sub-task exhibits degradation in performance for more than one model, suggesting that CoT prompting is broadly beneficial and rarely harmful within this domain.

The overall accuracy and robustness metrics also trend upward with CoT, reinforcing its value not only in structured reasoning but also in enhancing the model's resilience to perspective-related perturbations. For instance, the average gain in P&R Overall Accuracy is +5.59%, and in Robustness is +6.63%, indicating that step-by-step reasoning contributes to more confident and stable outputs.

While the benefits are widespread, a few failures still emerge. In Appendix, we analyze three representative failure cases to better understand CoT's limitations. These include GPT-4o on Perspective Type Reasoning, and Gemini-2-flash on Line Relationship and Perspective Transformation Spotting.

Table 2: **Chain of Thought (CoT) prompting improves MLLM performance on perspective tasks.** Accuracy changes due to CoT prompting across perception and reasoning tasks.

| | Perspective Perception | | | | Perspective Reasoning | | | | | P'Percep & P'Reason | | | P'Robust |
|---|---|---|---|---|---|---|---|---|---|---|---|---|---|
| | VPP | CLP | VAP | LDP | PTR | LRR | OVR | PTS | VPC | P Acc | R Acc | Overall | Binary |
| GPT-4o | +2.56 | +11.38 | +4.32 | +2.81 | -0.66 | +5.30 | +4.55 | +3.76 | +1.75 | +5.27 | +2.94 | +3.97 | +0.00 |
| Gemini-1.5-flash | +3.21 | +3.25 | +7.41 | +8.77 | +24.59 | +11.92 | +1.95 | +4.69 | +12.28 | +5.66 | +11.09 | +8.67 | +4.72 |
| Gemini-2-flash | +4.49 | +14.63 | -0.62 | +0.35 | +7.43 | -1.99 | +11.04 | -0.94 | +2.63 | +4.71 | +3.63 | +4.11 | +15.18 |
| Average $\Delta$ | +3.42 | +9.76 | +3.70 | +3.98 | +10.45 | +5.08 | +5.84 | +2.50 | +5.56 | +5.21 | +5.89 | +5.59 | +6.63 |

Overall, our findings suggest that while CoT prompting is not a silver bullet, it provides meaningful and reliable improvements in most perspective tasks. This points toward the promise of integrating structured reasoning strategies with visual understanding, especially for tasks where spatial interpretation and viewpoint deduction are required.

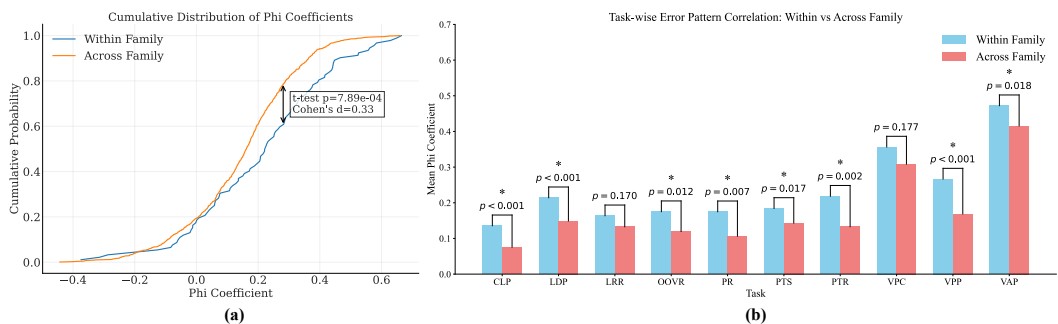

Figure 7: Error pattern analysis across model families: (a) Cumulative distribution of phi coefficients shows significantly higher correlations within families than across families (Cohen's $d = 0.33$, $p < 0.001$). (b) Task-wise breakdown reveals perception tasks (**VAP**, **CLP**) exhibit the strongest family-specific patterns, while reasoning tasks (**VPC**, **LRR**) show weaker family effects.

> **Finding 3.** Error pattern analysis reveals that while architectural/training choices strongly influence perspective perception biases, some spatial reasoning challenges present consistent difficulties across all model families.

Error correlations reveal that model architecture/training strongly influences perspective understanding failure modes. Fig. 7a demonstrates models from the same family exhibit significantly more similar error patterns than models from different families (Cohen's $d = 0.33$, $p < 0.001$), indicating architectural biases systematically affect perspective interpretation. The task-wise analysis in Fig. 7b reveals this family effect varies markedly across the perspective hierarchy: low-level perception tasks show the strongest architecture/training-specific biases, with **VAP** and **CLP** exhibiting the largest within-family versus across-family differences ($p < 0.001$). Notably, some tasks maintain relatively high correlation coefficients even in cross-family comparisons, particularly for **VAP** (0.41) and **VPC** (0.31). This suggests certain perspective challenges present universal difficulties that transcend architectural/training differences, especially tasks requiring complex spatial judgment (**VAP**) or precise counting of geometric features (**VPC**). In contrast, tasks like **CLP** show much larger gaps between within-family and cross-family correlations, indicating these capabilities are more sensitive to architectural design or training choices. These patterns reveal that while architecture significantly shapes perspective understanding biases, some fundamental spatial reasoning challenges remain consistently difficult across model designs.

## 4 Related Work

### 4.1 Perspective Understanding

Perspective is a cornerstone of visual realism, dictating how objects in a 2D image are perceived as three-dimensional. The theory of perspective can be traced back to Renaissance art, where principles such as VPs and HLs were formalized [Elkins, 1994, Haley, 2018]. In computer graphics and vision, perspective projection ensures that parallel lines in the real world converge at a VP on the image plane [Hartley and Zisserman, 2003]. Multiple VPs, depending on the orientation of objects, define

1-point, 2-point, or 3-point perspectives. Efficient and accurate VP detection has been a critical area of research, facilitating tasks like scene reconstruction [Lee et al., 2009, Hedau et al., 2009] and camera calibration [Zhang, 2000]. Techniques such as the Hough Transform [Duda and Hart, 1972] and its extensions [Candès et al., 2011] enable robust line detection, while Gaussian sphere mapping [Barnard, 1983] provides a framework for detecting intersections representing VPs. Classical methods often detect VPs through line segment intersections [Quan and Mohr, 1989, Lutton et al., 1994], followed by clustering approaches [McLean and Kotturi, 1995] or specialized voting schemes [Gamba et al., 1996]. Recent works leverage deep learning, with methods like NeurVPS [Zhou et al., 2019] that employ conic convolution operators and the Deep Hough Transform [Lin et al., 2022] to improve accuracy in VP detection across diverse datasets.

## 4.2 Evaluation Benchmarks for MLLMs

With the rapid advancement of MLLMs [Fei et al., 2024], numerous benchmarks have emerged to systematically evaluate diverse capabilities [Li et al., 2025b]. These benchmarks generally assess two dimensions: text-centric evaluations measuring commonsense knowledge and reasoning (MMMU [Yue et al., 2024], NaturalBench [Li et al., 2024a]), and vision-centric assessments focusing on perception and robustness (MMBench [Liu et al., 2024c], MME [Fu et al., 2024], Grit [Gupta et al., 2022]). Specialized visual tasks are evaluated through benchmarks for spatial relationship comprehension (SEED-Bench [Li et al., 2023a], MM-Vet [Yu et al., 2023]), chart understanding (MMSTAR [Chen et al., 2024a], MuirBench [Wang et al., 2024a]), visual grounding (Flickr30k [Plummer et al., 2015], TRIG [Li et al., 2025a]), and hallucination detection (POPE [Li et al., 2023b], HallusionBench [Guan et al., 2024]). Common evaluation approaches include image captioning [Lin et al., 2014, Onoe et al., 2024], Visual Question Answering [Antol et al., 2015, Marino et al., 2019, Mathew et al., 2020], and visual reasoning [Johnson et al., 2017, Suhr et al., 2017, Hua et al., 2025]. However, while certain benchmarks incorporate deeper assessments of perspective understanding remains limited [Thrush et al., 2022, Hua et al., 2024].

## 5 Conclusion

In this work, we introduce MMPerspective, the first benchmark to systematically evaluate perspective understanding in MLLMs. Through 10 tasks across perception, reasoning, and robustness, we reveal that while current models demonstrate basic geometric awareness, they fall short in compositional reasoning and maintaining consistency under perspective-preserving transformations. Our large-scale evaluation of 43 models uncovers clear performance trends and architectural limitations, pointing to the need for stronger spatial priors and geometry-aware design. MMPerspective provides a foundation for diagnosing perspective-related weaknesses and guiding the development of more spatially grounded vision-language systems.

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
