# Appendix

## A    Task Definitions

Table 3 outlines the and reasoning tasks included in the MMPerspective benchmark. Sample cases and representative questions are included to illustrate the task format and input style. We also show examples of perspective-invariant image operations for robustness evaluation in Figure 17, including cropping, masking, flipping, and rotation.

Table 3: Task and question definition in MMPerspective.

| | Task | # | Sample Case | Description | Sample Questions |
|---|---|---|---|---|---|
| **Perspective Perception** | Vanishing Point Perception (**VPP**) | 156 | Figure 8 | Identify the region that contains the vanishing point in the image. | Where is the vanishing point in this image? |
| | Critical Line Perception (**CLP**) | 123 | Figure 9 | Determine which of the highlighted lines is the horizon line. | Which line highlighted in the image is the Horizon Line? |
| | View Angle Perception (**VAP**) | 162 | Figure 10 | Infer the camera's line of sight direction from spatial cues. | What direction is the Line of Sight in this image? |
| | Lens Distortion Perception (**LDP**) | 285 | Figure 11 | Identify the region without curved-line distortion in the image. | Which region shows no curved-line distortion? |
| **Perspective Reasoning** | Perspective Type Reasoning (**PTR**) | 606 | Figure 12 | Classify the perspective type used in the image (e.g., one-point, two-point). | What is the perspective type of this image? |
| | Line Relationship Reasoning (**LRR**) | 151 | Figure 13 | Determine the spatial relationship between two lines in 3D (e.g., parallel, perpendicular). | What is the relationship between these two highlighted lines in the 3D space? |
| | Perspective Transformation Spotting (**PTS**) | 213 | Figure 14 | Identify the change in perspective type between two images. | What changes occur from the left image to the right image? |
| | Vanishing Point Counting (**VPC**) | 114 | Figure 15 | Count the number of vanishing points present in the image. | How many vanishing points can you identify within the image? |
| | Out-of-View Reasoning (**OVR**) | 308 | Figure 16 | Infer the quadrant location of an unseen vanishing point based on scene geometry. | In which quadrant might the vanishing point be located? |

**Vanishing Point Perception**

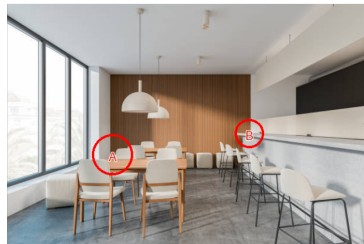

**Where is the vanishing point? Select from the following choices.**

(A) Circle A      (B) Circle B
(C) Both A and B      (D) None of the above

Answer: D

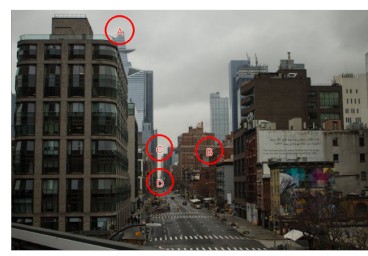

**Where is the vanishing point? Select from the following choices.**

(A) Circle A      (B) Circle B
(C) Circle C      (D) Circle D

Answer: A

Figure 8: Examples of Vanishing Point Perception.

**Critical Line Perception**

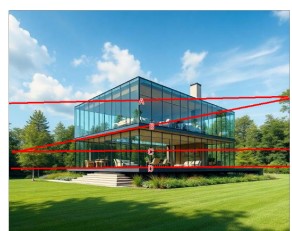

**Which line highlighted in the image aligns with the Line of Sight? Select from the following choices.**

(A) Line A      (B) Line B
(C) Line C      (D) Line D

Answer: D

**Which line highlighted in the image aligns with the Line of Sight? Select from the following choices.**

(A) Line A      (B) Line B
(C) Line C      (D) None of the above

Answer: D

Figure 9: Examples of Critical Line Perception.

**View Angle Perception**

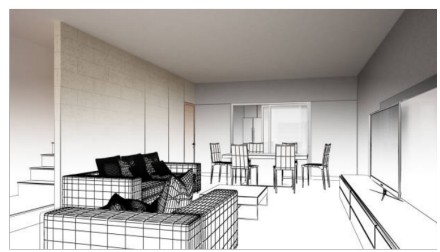

**What direction is the Line of Sight in this image? Select from the following choices.**

(A) Upward      (B) Downward
(C) Horizontal      (D) Unable to determine

Answer: C

**What direction is the Line of Sight in this image? Select from the following choices.**

(A) Upward      (B) Downward
(C) Horizontal      (D) Unable to determine

Answer: A

Figure 10: Examples of View Angle Perception.

**Lens Distortion Perception**

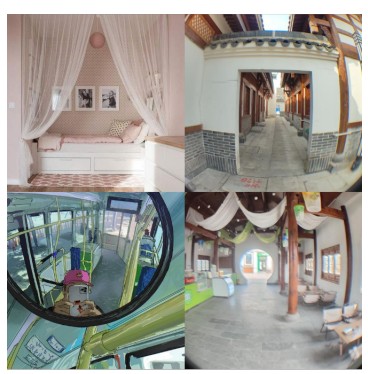
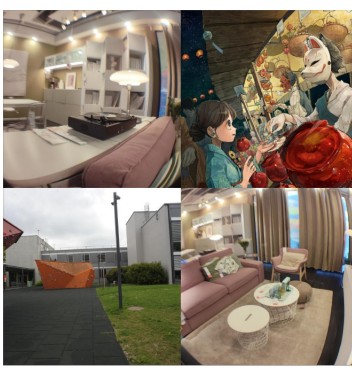

**Which region shows no curved-line distortion? Select from the following choices.**

(A) Top left         (B) Top right
(C) Bottom left        (D) Bottom right

Answer: A

**Which region shows no curved-line distortion? Select from the following choices.**

(A) Top left         (B) Top right
(C) Bottom left        (D) Bottom right

Answer: C

Figure 11: Examples of Line Relationship Reasoning.

**Perspective Type Reasoning**

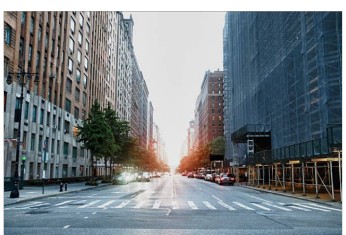

**What is the perspective type of this image? Select from the following choices.**

(A) One-point perspective   (B) Two-point perspective
(B) Three-point perspective  (D) Non-linear perspective

Answer: A

**What is the perspective type of this image? Select from the following choices.**

(A) One-point perspective   (B) Two-point perspective
(B) Three-point perspective  (D) Non-linear perspective

Answer: C

Figure 12: Examples of Perspective Type Reasoning.

**Line Relationship Reasoning**

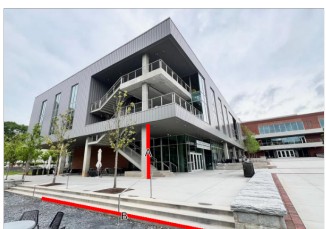

What is the relationship between these two highlighted lines in the 3D space as shown in the image? Select from the following choices.

(A) Intersecting            (B) Perpendicular
(C) Both A and B         (D) None of the above

Answer: B

What is the relationship between these two highlighted lines in the 3D space as shown in the image? Select from the following choices.

(A) Intersecting            (B) Perpendicular
(C) Both A and B         (D) Neither A nor B

Answer: C

Figure 13: Examples of Line Relationship Reasoning.

**Perspective Transformation Spotting**

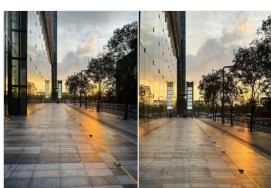

What changes occur from the left image to the right image? Select from the following choices.

(A) The image changes from a 1-point perspective to a 2-point perspective.
(B) The image changes from a 1-point perspective to a 3-point perspective.
(C) The image changes from a 3-point perspective to a 1-point perspective.
(D) The perspective type does not change.

Answer: D

What changes occur from the left image to the right image? Select from the following choices.

(A) The perspective type does not change.
(B) The image changes from a 2-point perspective to a 3-point perspective.
(C) The image changes from a 3-point perspective to a 2-point perspective.
(D) The image changes from a 1-point perspective to a 3-point perspective.

Answer: B

Figure 14: Examples of Perspective Transformation Spotting.

**Vanishing Point Counting**

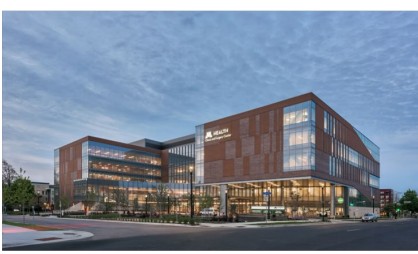

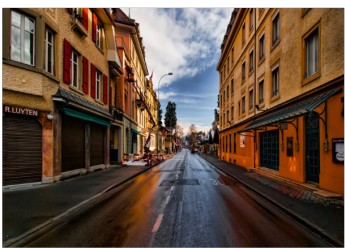

How many vanishing points can you identify WITHIN the image? Select from the following choices.

(A) 0            (B) 1
(C) 2            (D) 3

Answer: A

How many vanishing points can you identify WITHIN the image? Select from the following choices.

(A) 0            (B) 1
(C) 2            (D) 3

Answer: B

Figure 15: Examples of Vanishing Point Counting.

**Out-of-View Reasoning**

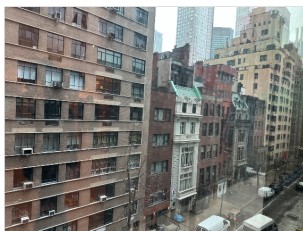

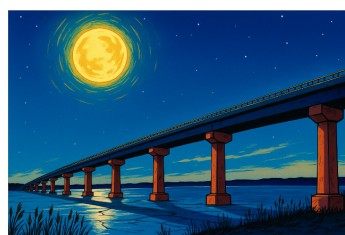

**In which quadrant might the vanishing point be located? Select from the following choices.**

(A) Bottom left        (B) Top right
(C) Bottom left and Bottom right    (D) Bottom right

Answer: D

**In which quadrant might the vanishing point be located? Select from the following choices.**

(A) Top left        (B) Bottom left
(C) Bottom right      (D) Top right

Answer: B

Figure 16: Examples of Out-of-View Reasoning.

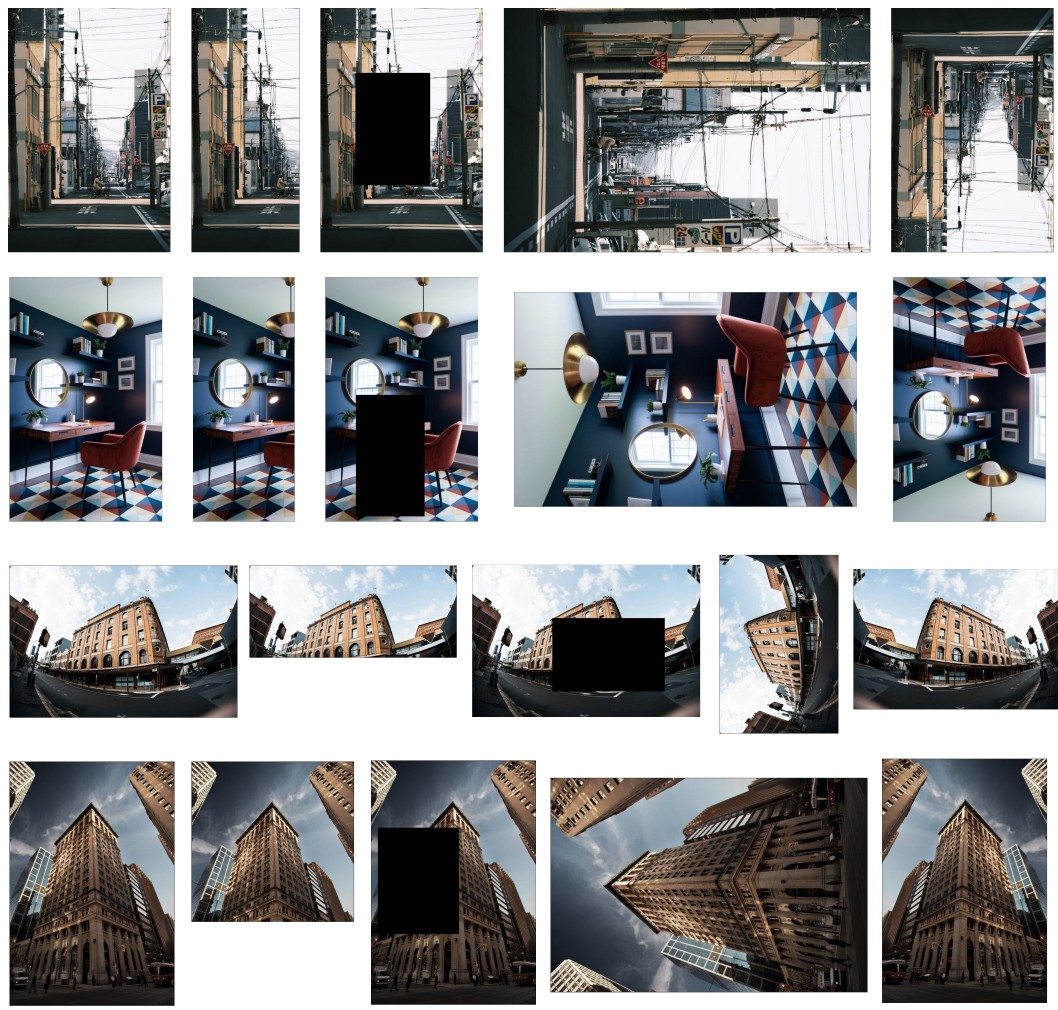

Figure 17: Examples of Perspective-Invariant Image Operations for Robustness Evaluation.

## B    More Terminology of Perspective

Figure 18 illustrates the key distinction between the Line of Sight (LS) and Horizon Line (HL) in perspective drawing. HL represents the viewer's eye level, while LS indicates the exact direction the viewer is looking. When LS is parallel to the ground, it aligns with HL, resulting in a typical 2-point perspective with verticals remaining straight. But when LS tilts upward or downward, it separates from HL, introducing vertical convergence and shifting the drawing into 3-point perspective. Importantly, the relative position of LS and HL also determines the view angle. If LS is above HL, the viewer is looking up (upward view); if it's below, the viewer is looking down (downward view). This shift changes what parts of an object are emphasized, more base or more top, and impacts how space is perceived.

## C    More Visualization

### C.1    Model Size & Performance for Each Task

In Figure 20 to 28, we present heatmaps for the 10 tasks in our **Perspective Perception**, **Perspective Reasoning**. The figures show the correlations between the sizes of **model parameters** and the metrics. Deeper color represents better performance. Each row represents a model family with the sizes growing from small to large. Most tasks clearly exhibit the correlation between model sizes and performance, i.e., larger model leads to higher metrics. However, Figure 27 shows that models with median size have better performance than smaller and larger models in **Perspective Transformation Spotting (PTS)**. Moreover, in **Vanishing Point Counting (VPC)**, we observe a reversed correlation where larger models lead to worse performance.

### C.2    Effect of Chain-of-Thought

Figures 29 to 32 are examples that demonstrate how Chain-of-Thought (CoT) can generally enhance the model's performance.

Despite the general enhancement, a few failures still emerge. Figures 33, 34, and 35 show three representative failure cases, including GPT-4o on Perspective Type Reasoning, and Gemini-2-flash on Line Relationship and Perspective Transformation Spotting. In these three cases, we all observed that the models made direct factual errors when analyzing the information in the images, rather than logical errors during the CoT process. This indicates that what limits the performance of the model is the ability to understand images.

### C.3    Performance for Each Model Family

Figure 36 and Figure 37 show task performance across various models within the same model families. Generally, models that are larger usually excel in most tasks.

### C.4    Question Difficulty Distribution

Figure 38 presents the question difficulty distribution based on average model accuracy. Each question is categorized into four difficulty levels, *Easy*, *Medium*, *Hard*, and *Super Hard*, based on the proportion of models that answered it correctly. The top two charts show the overall and type-level distributions, while the bottom figure provides a fine-grained view across tasks.

## D    Annotation

### D.1    Annotation Tool

We develop a dedicated annotation tool (see Figure 39) to support the systematic construction of multiple-choice questions in our benchmark. Designed specifically for perspective understanding, the tool enables annotators to load image pairs, formulate perspective-related questions, and select answers from a predefined list of geometric transformations (e.g., "1-point to 3-point perspective", "2-point to 1-point perspective"). This standardization ensures consistent labeling across the dataset. The

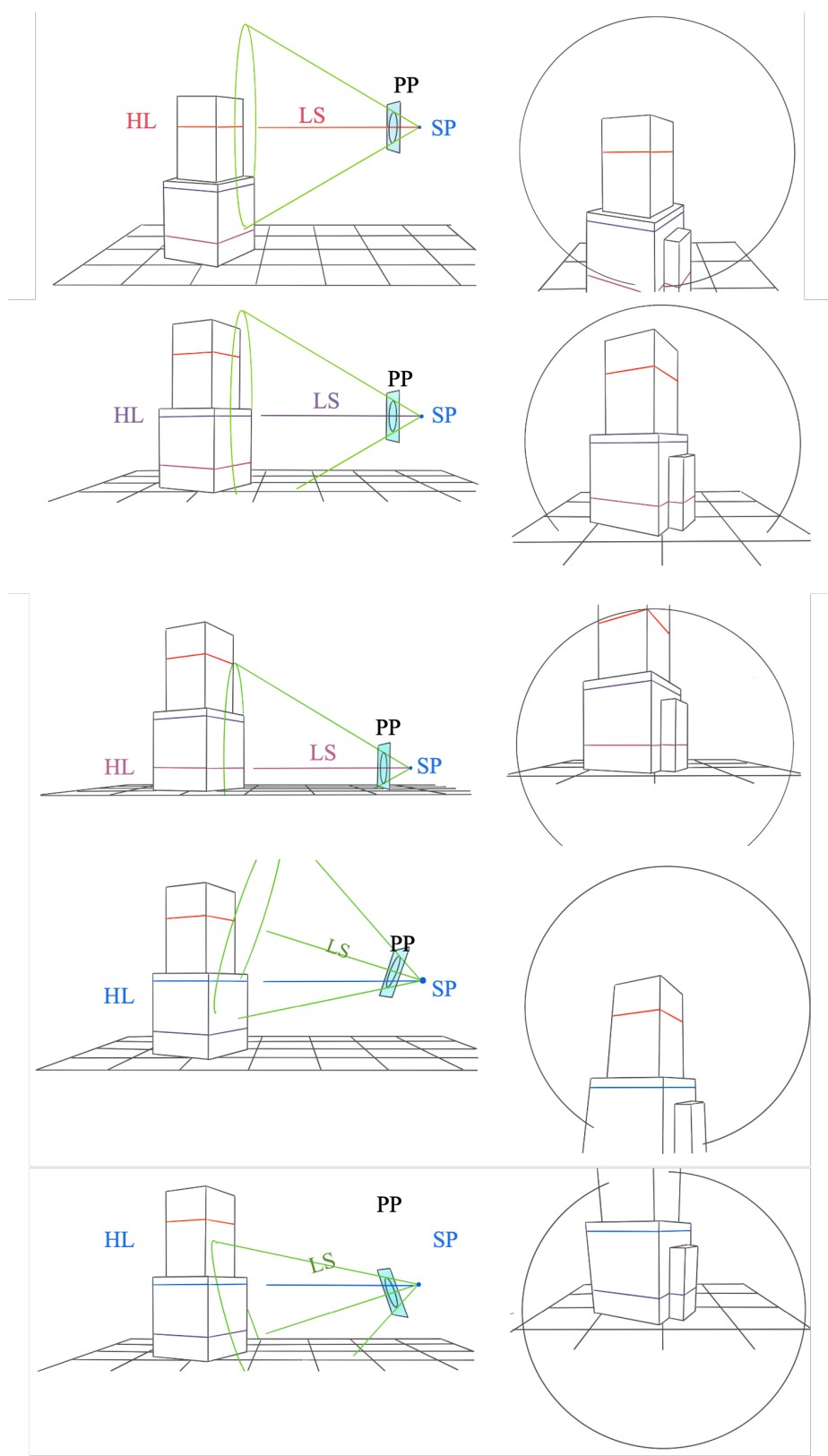

Figure 18: The relationship between Station Point (SP), Picture Plane (PP), Line of Sight (LS), and Horizon Line (HL) in perspective drawing. They demonstrate how viewing objects from different heights and angles affects spatial representation, emphasizing the critical distinction between LS and HL for accurate perspective construction. Figures are adapted from [Robertson and Bertling, 2013].

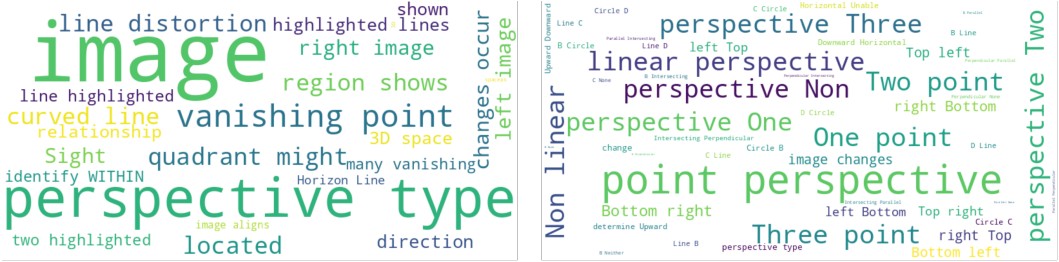

Figure 19: Word clouds of questions (left) and answer choices (right) in the MMPerspective Benchmark, illustrating the distribution of key terms related to perspective understanding.

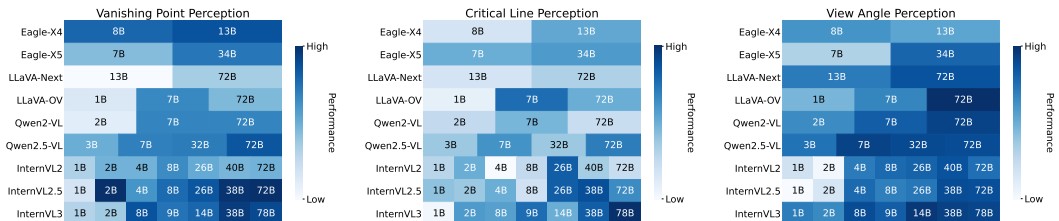

Figure 20: The heatmap for Vanishing Point Perception.

Figure 21: The heatmap for Critical Line Perception.

Figure 22: The heatmap for View Angle Perception.

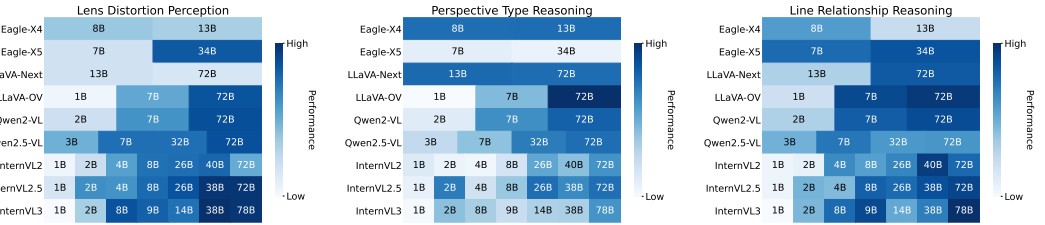

Figure 23: The heatmap for Lens Distortion Perception.

Figure 24: The heatmap for Perspective Type Reasoning.

Figure 25: The heatmap for Line Relationship Reasoning.

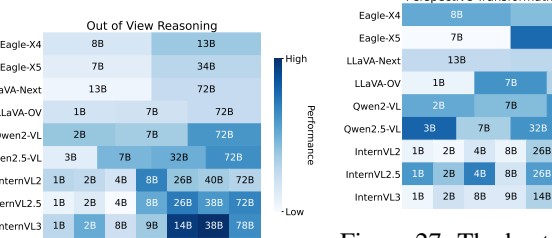
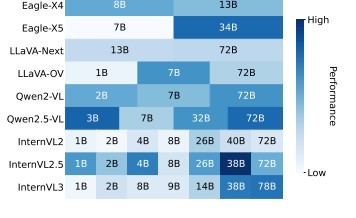
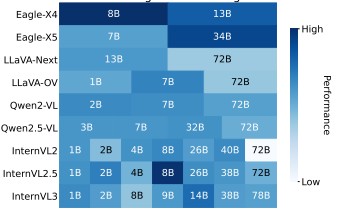

Figure 26: The heatmap for Out of View Reasoning.

Figure 27: The heatmap for Perspective Transformation Spotting.

Figure 28: The heatmap for Vanishing Point Counting.

interface integrates a suite of carefully designed features to facilitate precise annotation. Annotators can draw lines and circles to mark vanishing directions, orthogonal structures, or other relevant cues. Adjustable line width, zoom controls, and undo/redo functionality support detailed inspection and flexible editing. The tool also provides step-wise navigation through image sets and supports saving both visual annotations and structured Q&A data. By tailoring the design to the specific needs of perspective-based reasoning, the tool enables the efficient generation of high-quality, semantically grounded tasks. It plays a central role in ensuring the accuracy, consistency, and scalability of our benchmark construction.

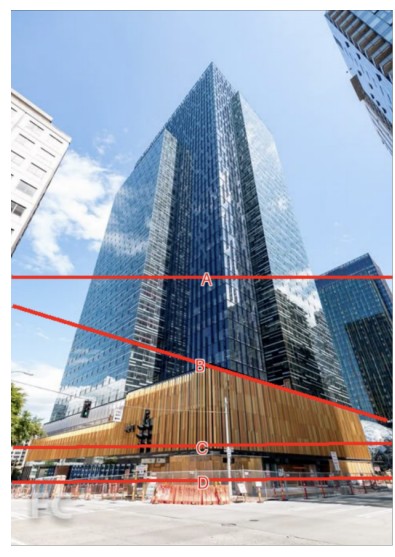

**Which line highlighted in the image aligns with the Line of Sight? Select from the following choices.**

(A) Line A (B) Line B
(C) Line C (D) Line D

Answer: A

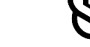

**GPT-4o w/o CoT: B**

**GPT-4o w/ CoT:**
To determine which line aligns with the line of sight to the building:
1. Understanding Line of Sight: It is a straight line from the observer's eye to the target object, usually parallel to the ground.
2. Analyzing the Lines: Line A is horizontal and level with the background and the horizon. Line B is angled upward. Line C is angled downward. Line D is also angled downward but close to the ground. The line of sight, being horizontal and parallel to the horizon, is best represented by a straight, level line. Therefore, the line that aligns with the line of sight is Line A. The correct option is **A**.

Figure 29: Examples of Chain-of-Thought Reasoning.

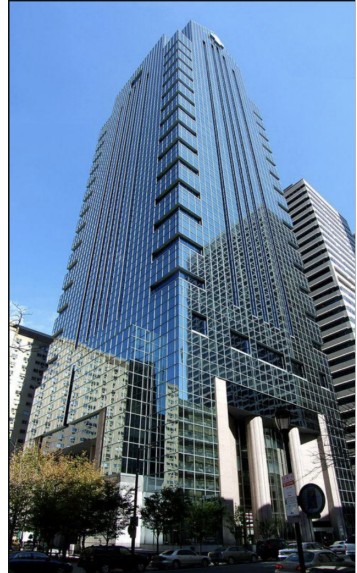

**What is the perspective type of this image? Select from the following choices.**
(A) One-point perspective
(B) Two-point perspective
(C) Three-point perspective
(D) Non-linear perspective

Answer: C

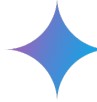

**Gemini-2-flash w/o CoT: B**

**Gemini-2-flash w/ CoT:**
The image shows a tall building. Let's analyze the vanishing points. One-point perspective means all lines converge to a single point. This isn't the case, as the vertical lines are mostly parallel to each other. Two-point perspective means there are two vanishing points. In the image, the horizontal lines on both sides of the building converge to a vanishing point to the left and to the right. However, the vertical lines also seem to converge to a point above. Three-point perspective means there are three vanishing points. This usually involves convergence toward a point above or below, in addition to the two horizontal vanishing points. In this case, it is clear that all three axes converge to three vanishing points. Non-linear perspective does not use vanishing points, and parallel lines may appear curved. This is not the case here. Since the vertical lines also converge, this is three-point perspective. The correct answer is **C**.

Figure 30: Examples of Chain-of-Thought Reasoning.

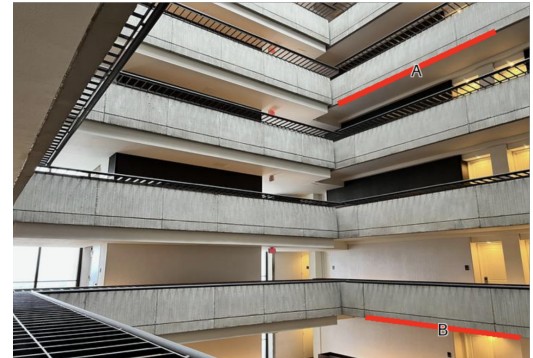

**What is the relationship between these two highlighted lines in the 3D space as shown in the image? Select from the following choices.**
(A) Intersecting (B) Parallel (C) Perpendicular (D) None of the above

Answer: B

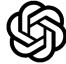

**GPT-4o w/o CoT: A**

**GPT-4o w/ CoT:**
To determine the relationship between the two highlighted lines A and B in 3D space:
1. Position: They are located on different levels or floors of a building, so they are not in the same plane.
2. Direction: Both lines follow the direction of the walkways on their respective levels and run along the edges.
3. Intersection: Since they are on separate planes, they do not intersect.
4. Parallelism: The lines appear to maintain a consistent distance apart, suggesting they would never meet if extended.
5. Perpendicularity: They do not form a 90-degree angle with each other.
Based on this, the lines are parallel since they run in the same direction and do not intersect. The correct choice is **B**. Parallel.

Figure 31: Examples of Chain-of-Thought Reasoning.

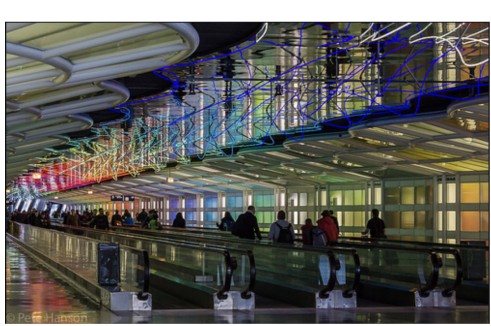

**In which quadrant might the vanishing point be located? Select from the following choices.**
(A) Bottom left and Bottom right (B) Top left (C) Bottom right (D) Bottom left

Answer: D

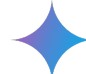

**Gemini-2-flash w/o CoT: A**

**Gemini-2-flash w/ CoT:**
The vanishing point is where parallel lines appear to converge in a perspective drawing. In this image, the hallway's parallel lines angle toward the left and downward. Following their direction, they appear to meet in the bottom left area of the image.

Therefore, the vanishing point is likely in the bottom left quadrant.

The correct answer is **D**.

Figure 32: Examples of Chain-of-Thought Reasoning.

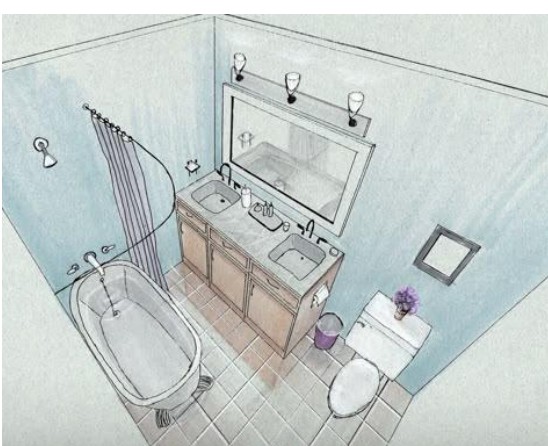

**What is the perspective type of this image? Select from the following choices.**

(A) One-point perspective
(B) Two-point perspective
(C) Three-point perspective
(D) Non-linear perspective

------------------------------------------------
**Answer: C**

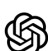

**GPT-4o w/ CoT: C**

**GPT-4o w/ CoT:**

To determine the perspective type:

1. **One-Point Perspective:** This features a single vanishing point on the horizon line where all lines converge. It's usually used for views looking directly at a flat surface.

2. **Two-Point Perspective:** This has two vanishing points on the horizon line. It's often used to depict a corner where two sets of parallel lines converge toward two vanishing points.

3. **Three-Point Perspective:** This incorporates a third vanishing point either above or below the horizon line and is used for extreme perspectives, such as looking up at or down from a high point.

4. **Non-Linear Perspective:** This doesn't follow linear perspective rules and can have distortions or different types of projections.

**Analysis of the Image:**
- The image shows a view from above, looking down into a bathroom.
- The lines appear to converge towards two vanishing points along the horizon, indicative of a two-point perspective.

This image employs a two-point perspective since it displays converging lines towards two distinct vanishing points.

**Conclusion:** The correct option is (**B**) Two-point perspective.

Figure 33: Examples of GPT-4o with Chain-of-Thought Reasoning in Perspective Type Reasoning.

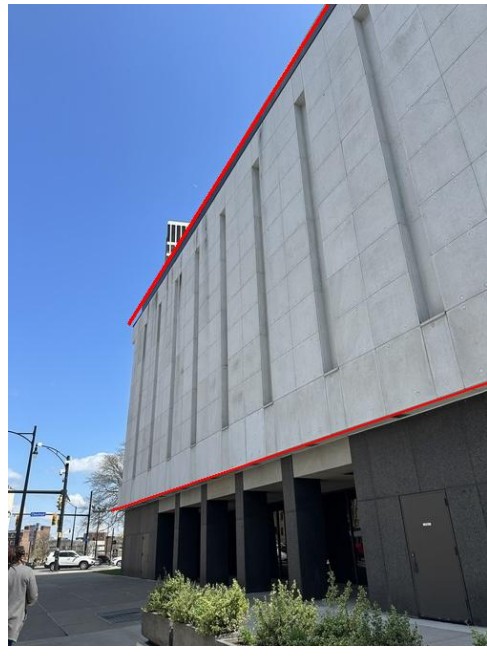

**What is the relationship between these two highlighted lines in the 3D space as shown in the image? Select from the following choices.**

(A) Perpendicular
(B) Intersecting
(C) Both A and B
(D) Parallel

----------------------------------------------------

**Answer: D**

**Gemini-2-flash w/ CoT: D** 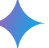

**Gemini-2-flash w/ CoT:**

The two highlighted lines are the top and bottom edges of the building. From the image, it appears that the lines would meet at some point if extended.
Therefore, the lines are intersecting. Since the lines are not at a 90-degree angle, they are not perpendicular.

(B)

Figure 34: Examples of Gemini-2-flash with Chain-of-Thought Reasoning in Perspective Transformation Spotting.

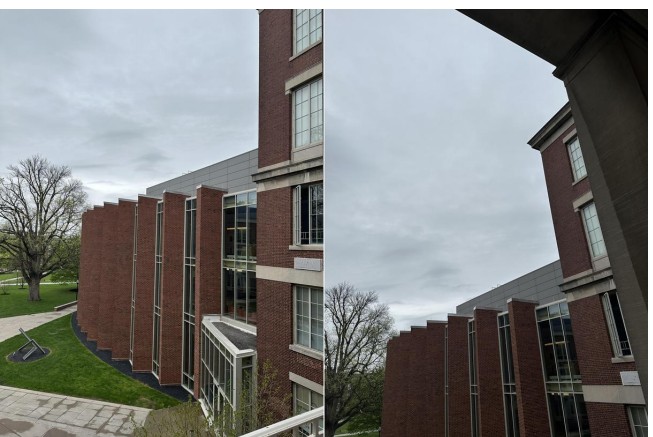

**What changes occur from the left image to the right image? Select from the following choices.**

(A) The image changes from a 1-point perspective to a 2-point perspective.
(B) The image changes from a 1-point perspective to a 3-point perspective.
(C) The image changes from a 2-point perspective to a 3-point perspective.
(D) The image changes from a 3-point perspective to a 2-point perspective.

----------------------------------------------------

**Answer: C**

**Gemini-2-flash w/ CoT: C** 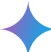

**Gemini-2-flash w/ CoT:**

Let's analyze the images to determine the type of perspective used in each.

In the left image, the building's edges appear to converge towards a single vanishing point that is far off to the right. This is characteristic of a 1-point perspective.

In the right image, the building's edges appear to converge towards two vanishing points. One is off towards the left, and one is towards the right. Thus, it is a 2-point perspective.

Therefore, the change is from a 1-point perspective to a 2-point perspective.

(A)

Figure 35: Examples of Gemini-2-flash with Chain-of-Thought Reasoning in Line Relationship Reasoning.

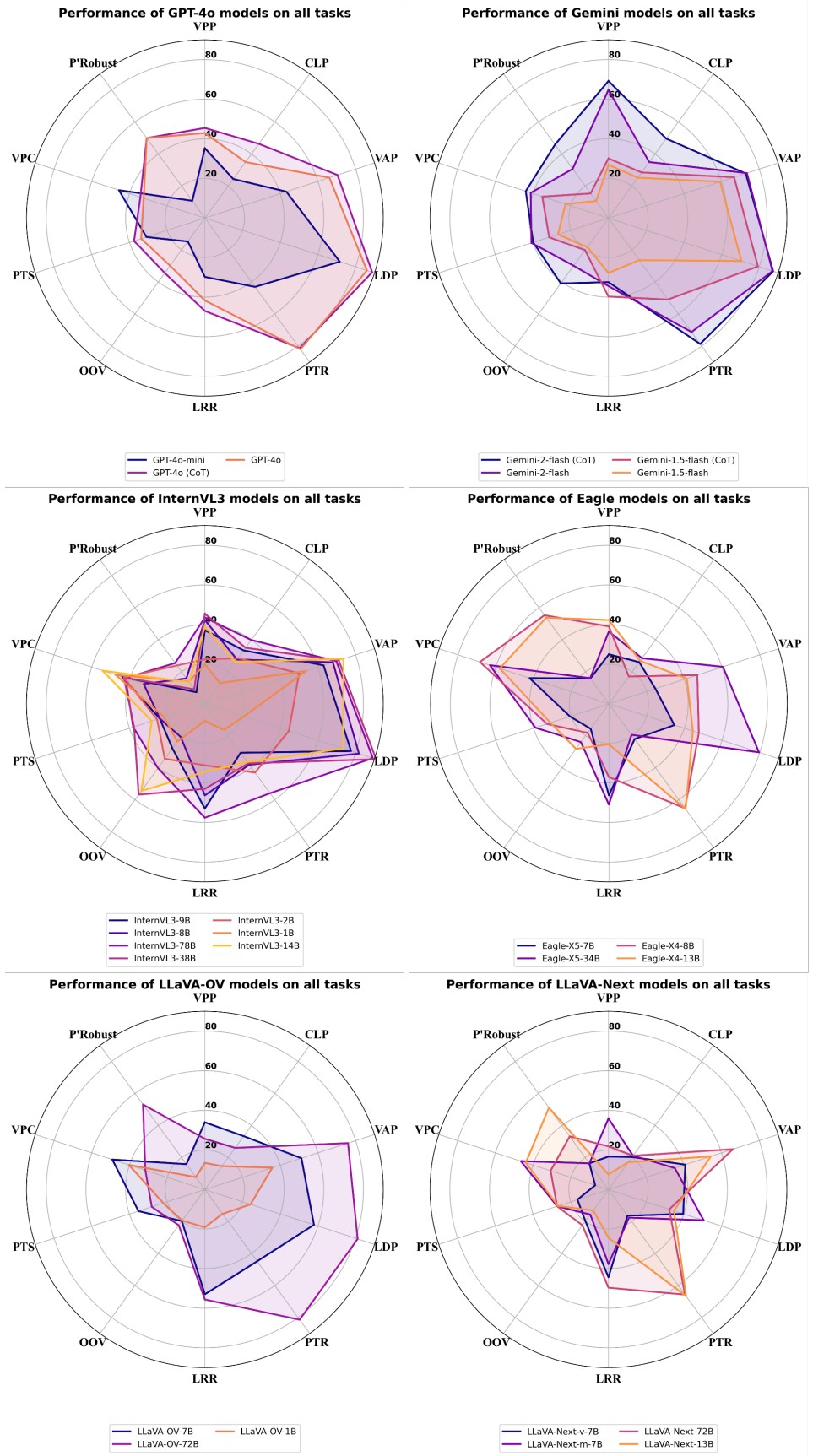

Figure 36: Task performance of models within each family (part 1).

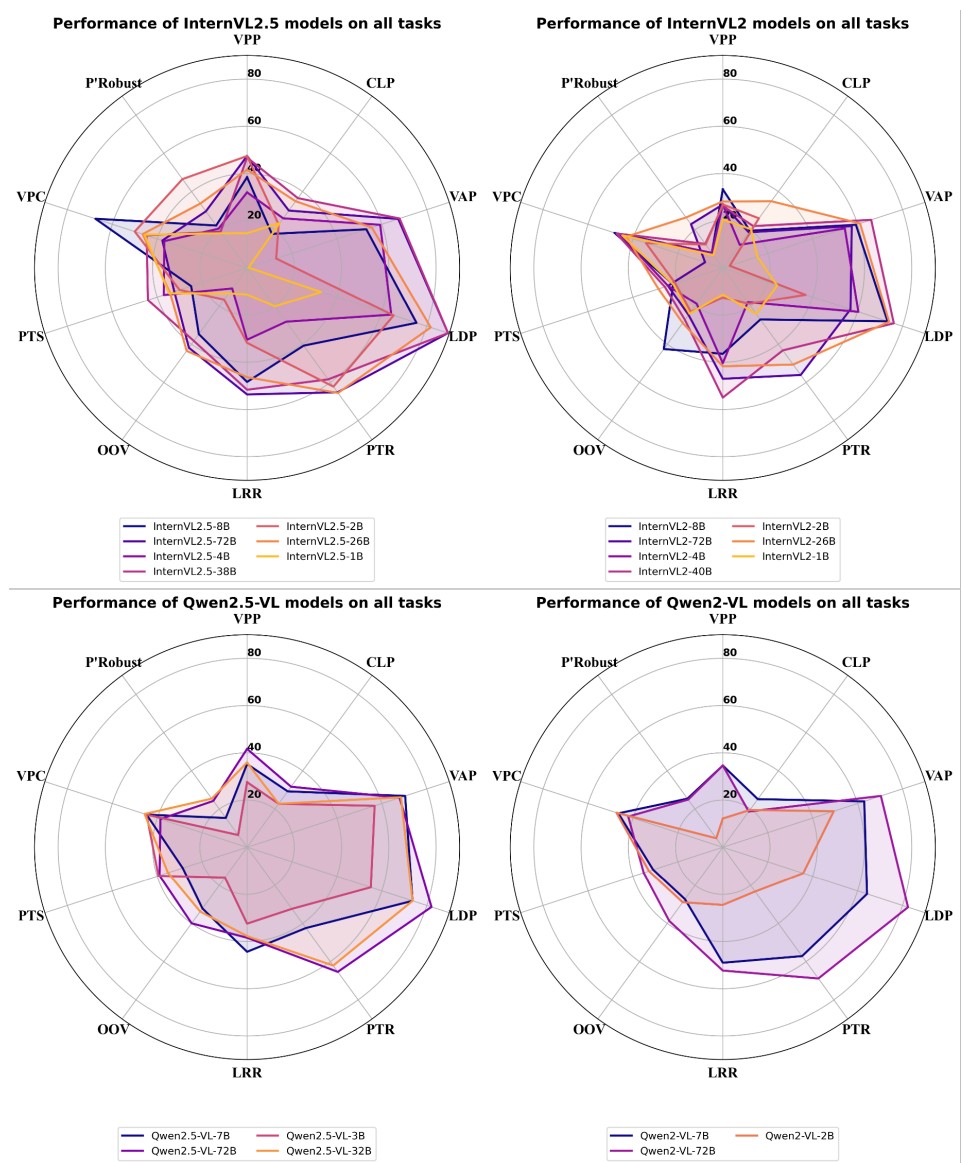

Figure 37: Task performance of models within each family (part 2).

## D.2 Annotator Background and Expertise

To further clarify, our annotation process involved a valuable collaboration between our internal research team and an external team of domain experts. Our internal team consists of 12 graduate students with backgrounds in computer vision, who received specific training on perspective principles using our custom annotation tool. Additionally, we partnered with a firm specializing in artistic perspective training. Their team of professional instructors, after learning about our project, generously contributed a portion of highly specialized annotations on a pro bono basis. This collaboration ensured our benchmark benefits from both technical computer vision oversight and deep, practical expertise in geometric perspective, guaranteeing a high quality of annotation.

## E More Results

We also have experiments to assess the effect of In-Context Learning (ICL). Our experiment followed a rigorous one-shot In-Context Learning paradigm. For each test question, we randomly sampled

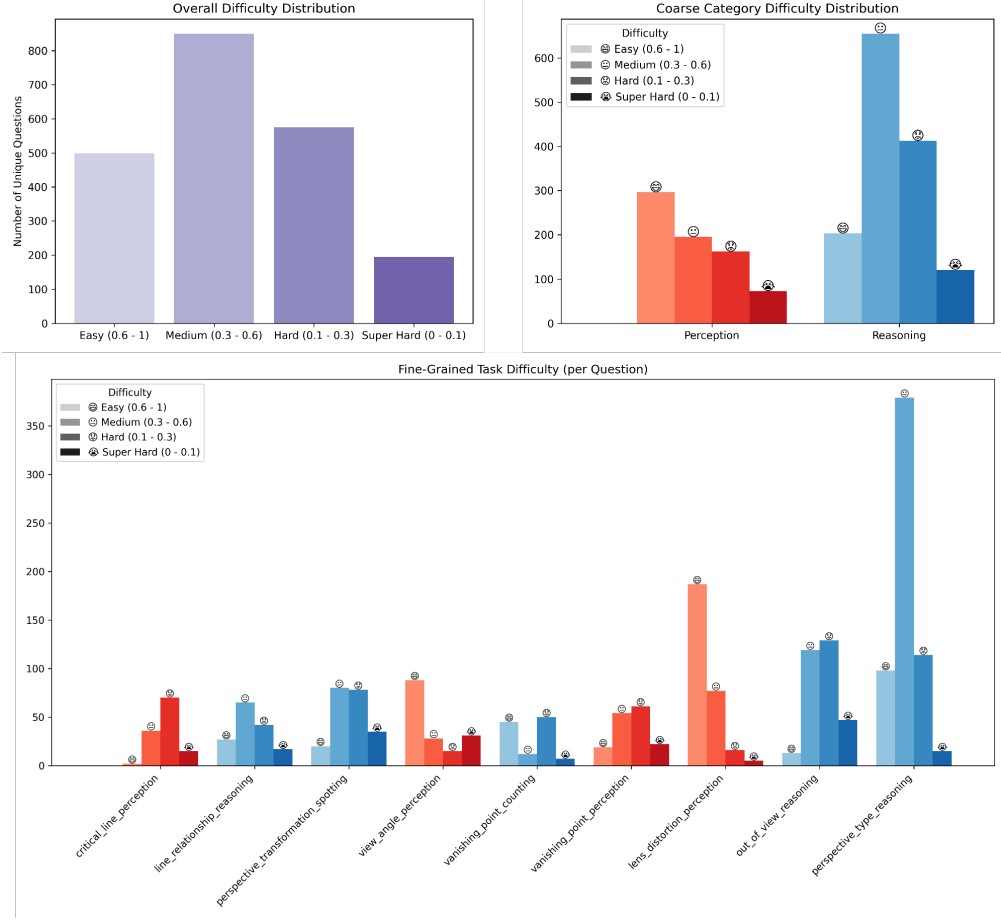

Figure 38: Distribution of question difficulty across task types.

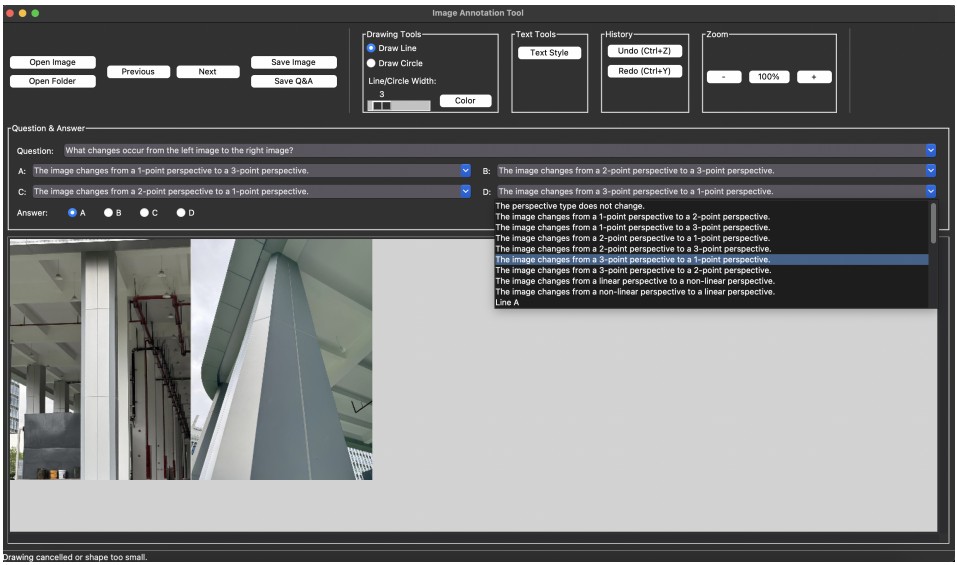

Figure 39: Annotation interface developed for constructing perspective-based multiple-choice questions. The tool integrates geometric drawing utilities, structured answer selection, and image navigation to support precise and consistent labeling.

another distinct image-question-answer pair from the same task category and prepended it to the prompt as an example. This approach primes the model with the task format without leaking the test answer. The results are shown in Table 4.

Table 4: The experiments to assess the effect of in-context learning (ICL).

| Settings | Perspective Perception | | | | Perspective Reasoning | | | | | P'Percep & P'Reason | | | Robustness | |
|---|---|---|---|---|---|---|---|---|---|---|---|---|---|---|
| | VPP | CLP | VAP | LDP | PTR | LRR | OVR | PTS | VPC | P Acc | R Acc | Overall | Graded | Binary |
| GPT-4o-mini | 35.3 | 24.4 | 43.2 | 71.6 | 43.1 | 29.8 | 14.6 | 31.0 | 45.6 | 43.6 | 32.8 | 37.6 | 28.7 | 10.8 |
| GPT-4o-mini (ICL) | 28.2 | 25.2 | 53.1 | 76.8 | 28.1 | 26.5 | 16.9 | 25.8 | 14.9 | 45.8 | 22.4 | 34.1 | 17.9 | 6.2 |
| GPT-4o | 42.9 | 35.0 | 66.0 | 86.0 | 82.0 | 41.7 | 29.9 | 33.8 | 32.5 | 57.5 | 44.0 | 50.0 | 71.9 | 49.9 |
| GPT-4o (ICL) | 55.1 | 43.1 | 71.6 | 92.6 | 80.7 | 51.7 | 40.9 | 42.3 | 39.5 | 65.6 | 51.0 | 58.3 | 72.2 | 53.5 |

Our analysis indicates that ICL's effectiveness may be tied to model scale. The one-shot example significantly increases the performance of the larger GPT-4o (increasing overall accuracy from 50.0% to 58.3%), but appears to be detrimental to the smaller GPT-4o-mini (decreasing from 37.6% to 34.1%). This divergence suggests that larger models may be better equipped to generalize from in-context examples for this task.

# F   Limitations

While MMPerspective provides a comprehensive benchmark for evaluating perspective understanding in MLLMs, several limitations should be acknowledged. First, the benchmark primarily focuses on static images and multiple-choice question answering (MCQA) formats, which may not fully capture the depth of spatial reasoning required in dynamic or open-ended tasks. Real-world applications often demand free-form generation, spatial manipulation, or multi-turn interactions that extend beyond our current evaluation scope. Our choice of the MCQA format was a deliberate design decision based on several considerations:

- **Objectivity and Scalability:** The MCQA format allows for automated, objective, and large-scale evaluation, avoiding the subjectivity and high cost associated with evaluating open-ended responses.
- **Controlled Probing:** This format enables us to precisely probe a model's understanding of specific geometric concepts (e.g., distinguishing between "two-point" and "three-point" perspective) without the noise from variations in natural language generation.
- **A Foundational First Step:** As the first benchmark in this domain, we believe that establishing a controlled and rigorous evaluation framework is a critical first step. It lays a solid foundation for future work that can build upon our benchmark with more complex, generative tasks.

Second, although we curated a diverse set of real and synthetic images, the dataset still exhibits a bias toward architectural and indoor scenes, which may limit generalizability to natural environments or abstract visual contexts. Third, despite our efforts to standardize evaluation, some tasks inevitably contain ambiguous visual cues, and model errors may stem from subjective interpretations rather than a lack of geometric understanding. Lastly, our benchmark assumes that all correct answers are equally accessible across models without considering differences in input modalities, prompting formats, or underlying vision-language alignment strategies. Future work could address these limitations by incorporating more open-ended tasks, expanding domain diversity, and developing adaptive evaluation protocols that account for model-specific reasoning pathways.