# OpenReview forum: "MMPerspective: Do MLLMs Understand Perspective? A Comprehensive Benchmark for Perspective Perception, Reasoning, and Robustness"
_NeurIPS.cc/2025/Datasets_and_Benchmarks_Track — NeurIPS 2025 Datasets and Benchmarks Track poster_

### Official Review · Reviewer_APxB · 2025-06-23

**Rating:** 5
**Confidence:** 3

**Summary:**

This paper presents a novel and compelling benchmark called MMPerspective that addresses a crucial challenge: understanding perspective through perception, reasoning, and robustness. Building on this benchmark, this paper evaluates representative MLLMs, offering valuable insights into current model limitations and bottlenecks.

**Dataset Code Accessibility:**

Yes

**Ethical Considerations:**

No, there are no or only very minor ethics concerns

**Final Justification:**

The authors responsed most of my concerns. Thus I raised my rate.

**Limitations Weaknesses:**

Since multiple-choice question answering is used for evaluation, how sensitive are the results to the order of answer choices? Specifically, if the options are randomly shuffled, do the models still perform consistently?

Many LLMs use sampling-based decoding strategies (e.g., temperature-based sampling). Was this randomness considered in the evaluation? In other words, how consistent are the model outputs across multiple runs on the same samples?

The purpose of the visualization in Figure 1 (right) is unclear. Refining this figure to better emphasize the core message would improve its effectiveness.

Line 90: The figure reference appears to be incorrect; it likely refers to Figure 3, not Figure 4.

Line 162: The benchmark filtering process lacks sufficient detail. What was the size of the raw dataset before filtering? How many samples were excluded? Additionally, it would be helpful to clarify whether the annotators were external contributors or part of the author team.

Regarding in-context learning: Was it explored in the evaluation? Including results with in-context examples could offer further insight into model capabilities and performance improvements.

**Strengths Contributions:**

The data collection process is clearly and thoroughly described, supporting the benchmark's quality and reproducibility.

The visualizations are clear and effective in conveying the key aspects of the benchmark.

The evaluation protocol is well-structured and complementary, providing a comprehensive assessment of model performance.

---

> ### Author Rebuttal · Authors · 2025-07-31
>
> Dear Reviewer APxB,
>
> Thank you for your meticulous review and your valuable feedback. We are pleased that you find our benchmark novel and compelling, and we appreciate your positive comments on our data collection process, visualizations, and evaluation protocol. Your specific and technical questions are crucial for improving the rigor and clarity of our work.
>
> Here is our point-by-point response to your questions:
>
> ### **Q1: Sensitivity to the order of answer choices.**
>
> To ensure fairness and eliminate potential positional bias, we have already randomly shuffled the answer choices for all questions during the dataset creation process. Therefore, the distribution of the correct answer (e.g., "two-point perspective") across options A, B, C, and D is balanced. We did not perform a second dynamic shuffling during evaluation, as we believe this initial randomization effectively mitigates the issue. We will explicitly state in the experimental setup (Section 3.1) of our final manuscript that the order of all answer choices was pre-randomized to ensure a fair evaluation.
>
> ### **Q2: Randomness in sampling-based decoding.**
>
> To ensure deterministic and fully reproducible results for all our experiments, we employed a greedy decoding strategy for all open-source models. For proprietary models accessed via API, we also used their deterministic decoding modes where available. This approach eliminates randomness from the decoding process, ensuring that a model's output for any given sample is consistent across multiple runs. We will add this clarification to Section 3.1 to explicitly state the deterministic decoding strategy used to ensure reproducibility.
>
> ### **Q3: The purpose of the visualization in Figure 1 (right).**
>
> The purpose of the right side of Figure 1 is to illustrate the workflow of our Perspective Robustness evaluation dimension. It shows how an original image is subjected to perspective-invariant transformations (e.g., flipping, cropping), and how we test whether a model maintains a consistent answer across all variants. We acknowledge that the current visualization may be complex. In the final version, we will revise this part of the figure to be simpler and clearer.
>
> ### **Q4: Incorrect figure reference in Line 90.**
> Yes, the reference should be to Figure 3, not Figure 4. We appreciate you catching this error. We will correct this typo in the final manuscript.
>
>
> ### **Q5: Lack of detail in the filtering process and annotators.**
>
> Thank you for the suggestion to improve transparency. We provided some details in the paper, but are happy to elaborate further.
>
> - **About dataset size and filtering**: We started with a raw dataset of approximately 3,500 candidate images. During the quality control phase, we excluded around 800 samples due to ambiguity, low quality, or irrelevance, resulting in the final benchmark of 2,711 instances.
>
> - **About annotator affiliation**: To further clarify, our annotation process involved a valuable collaboration between our internal research team and an external team of domain experts. Our internal team consists of 12 graduate students with backgrounds in computer vision, who received specific training on perspective principles using our custom annotation tool (described in Appendix D). Additionally, we partnered with a firm specializing in artistic perspective training. Their team of professional instructors, after learning about our project, generously contributed a portion of highly specialized annotations on a pro bono basis. This collaboration ensured our benchmark benefits from both technical computer vision oversight and deep, practical expertise in geometric perspective, guaranteeing a high quality of annotation.
>
> We will add these specific statistics and details to Section 2.3 Quality Control or the Appendix in the final version.
>
>
> ### **Q6: Exploring In-Context Learning (ICL).**
>
> While our current work focused on zero-shot and CoT settings, we agree that exploring ICL offers valuable insights. Inspired by your feedback, we conducted a new experiment to assess the effect of ICL.
>
> Our experiment followed a rigorous one-shot In-Context Learning paradigm. For each test question, we randomly sampled another distinct image-question-answer pair from the same task category and prepended it to the prompt as an example. This approach primes the model with the task format without leaking the test answer.
>
> Below are the updated results for the GPT-4o and GPT-4o-mini models:
>
> | Settings             | VPP   | CLP   | VAP   | LDP   | PTR   | LRR   | OVR   | PTS   | VPC   | P'Percep | P'Reason | Overall Acc | Graded P'Robust | Binary P'Robust |
> |----------------------|-------|-------|-------|-------|-------|-------|-------|-------|-------|-----------|-----------|--------------|------------------|------------------|
> | GPT-4o-mini          | 35.3  | 24.4  | 43.2  | 71.6  | 43.1  | 29.8  | 14.6  | 31.0  | 45.6  | 43.6      | 32.8      | 37.6         | 28.7             | 10.8             |
> | GPT-4o-mini (CoT)    | 32.7  | 25.2  | 47.5  | 44.2  | 55.9  | 41.7  | 24.7  | 39.0  | 34.2  | 39.3      | 43.1      | 41.8         | 41.9             | 9.3              |
> | GPT-4o-mini (ICL)    | 28.2  | 25.2  | 53.1  | 76.8  | 28.1  | 26.5  | 16.9  | 25.8  | 14.9  | 45.8      | 22.4      | 34.1         | 17.9             | 6.2              |
> | GPT-4o               | 42.9  | 35.0  | 66.0  | 86.0  | 82.0  | 41.7  | 29.9  | 33.8  | 32.5  | 57.5      | 44.0      | 50.0         | 71.9             | 49.9             |
> | GPT-4o (CoT)         | 45.5  | 46.3  | 70.4  | 23.2  | 79.5  | 42.4  | 34.7  | 38.5  | 34.2  | 62.7      | 46.9      | 54.0         | 69.4             | 44.4             |
> | GPT-4o (ICL)         | 55.1  | 43.1  | 71.6  | 92.6  | 80.7  | 51.7  | 40.9  | 42.3  | 39.5  | 65.6      | 51.0      | 58.3         | 72.2             | 53.5             |
>
> ---
>
> Our preliminary analysis indicates that ICL's effectiveness may be tied to model scale. The one-shot example significantly boosts the performance of the larger GPT-4o (increasing overall accuracy from 50.0% to 58.3%), but appears detrimental to the smaller GPT-4o-mini (decreasing from 37.6% to 34.1%). This divergence suggests larger models may be better equipped to generalize from in-context examples for this task. While based on a limited sample, these findings present a compelling direction for future investigation.
>
> We will include these results with analysis in the appendix of our final manuscript and will expand the experiment to include other models for a more comprehensive view.
>
> Thank you again for your thoughtful and detailed questions! We hope the response above effectively addresses your concerns.

---

> > ### Comment · Reviewer_APxB · 2025-08-05
> >
> > The authors responsed most of my concerns. Thus I raised my rate to accept.

---

### Official Review · Reviewer_abWa · 2025-07-03

**Rating:** 5
**Confidence:** 4

**Summary:**

This paper introduces MMPerspective, the first benchmark designed to systematically evaluate how well Multimodal Large Language Models (MLLMs) understand visual perspective. Through 10 tasks across three dimensions—Perception, Reasoning, and Robustness—the benchmark assesses 43 leading models. The study finds that while current models perform reasonably well at identifying basic perspective cues, their performance drops significantly when tasks require compositional reasoning or when images are perturbed, revealing deep limitations in their ability to internalize and apply fundamental geometric principles.

**Additional Feedback:**

The authors have done an excellent job of diagnosing the problem. A natural and valuable next step would be to build on these findings and propose a solution themselves. For instance, they could design a lightweight, geometry-aware module and demonstrate its effectiveness in improving existing models' performance on the MMPerspective benchmark. Such follow-up work would make their research contribution more complete, moving from "problem finding" to "problem solving."

**Dataset Code Accessibility:**

Yes

**Ethical Considerations:**

No, there are no or only very minor ethics concerns

**Final Justification:**

Author's rebuttal resolves my concerns, I'd like keep my rating as Accept

**Limitations Weaknesses:**

* Limited Evaluation Format: The benchmark relies entirely on static images and multiple-choice questions, which restricts its ability to assess a model's understanding of perspective in more complex scenarios, such as dynamic scenes, free-form text generation, or image generation.
* Potential Dataset Bias: The dataset is primarily focused on architectural and indoor scenes where linear perspective cues are most prominent. This may limit the generalizability of the findings to other visual domains like natural landscapes.
* Focus on Diagnosis over Solution: As a benchmark paper, its main contribution is to "identify the problem." It does not propose a new model or algorithm to directly address the identified weaknesses in perspective understanding, leaving solutions to future work.

**Strengths Contributions:**

* Pioneering and Important: The paper fills a critical gap in MLLM evaluation by shifting the focus from high-level semantics to the more fundamental and core ability of understanding visual geometry. This is a highly original contribution.
* Systematic and In-depth: The "Perception-Reasoning-Robustness" framework is cleverly designed and hierarchical, allowing for a deeper analysis of model capabilities than single-task evaluations. The robustness tests, in particular, are effective at distinguishing true understanding from superficial pattern matching.
* Comprehensive and Insightful: The evaluation covers a wide range of models, from small open-source ones to top-tier proprietary systems. The analysis is thorough, and its findings—such as the relationship between model size, performance, and robustness, and the effect of CoT prompting—provide valuable insights and future research directions for the community.

---

> ### Author Rebuttal · Authors · 2025-07-31
>
> Dear Reviewer abWa,
>
> We sincerely thank you for your insightful and highly positive review. We are immensely encouraged by your assessment of our work as "pioneering and important" and your recognition of our benchmark's systematic design, comprehensive evaluation, and insightful findings. We are honored to receive your recommendation for "Accept."
>
> The limitations you have pointed out are also very pertinent. Here is our point-by-point response to your concerns:
>
> ### **Q1 & Q2: Limited Evaluation Format and Potential Dataset Bias.**
>
> As you noted, our benchmark currently focuses on static images and a multiple-choice format, and is primarily composed of architectural and indoor scenes where linear perspective cues are most salient.
>
> We also acknowledged these deliberate design choices in our paper's limitations section (Appendix E). Our rationale for this controlled setup was to establish a clear, unambiguous, and quantifiable baseline for the complex and novel evaluation area of perspective understanding. Architectural and indoor scenes provide the most unequivocal geometric structures, which are essential for a first-of-its-kind, systematic assessment of a model's foundational geometric capabilities.
>
> In the final version, we will more explicitly articulate the reasoning behind these design choices and emphasize that they point to clear directions for future research. This includes expanding the evaluation to dynamic scenes, open-ended generation tasks, and more diverse visual domains like natural landscapes.
>
> ### **Q3 & Additional Feedback: Focus on Diagnosis over Solution.**
>
> You have accurately characterized the core contribution of our work as "problem finding." As a benchmark paper, our primary goal was to provide the community with a reliable diagnostic tool to reveal the systematic shortcomings of even state-of-the-art models in core visual geometry understanding. We firmly believe that a precise "diagnosis" is a necessary prerequisite for effective "problem solving."
> We are very grateful for your valuable suggestions for follow-up work. The idea of "designing a lightweight, geometry-aware module" to improve existing models is an excellent and promising research direction. This is precisely the kind of research we hope our MMPerspective benchmark will inspire and facilitate.
>
> In our conclusion or future work section, we will more clearly articulate how our benchmark paves the way for "problem-solving" research. We may cite the direction you suggested as a concrete example of how our diagnostic tool can spur innovative solutions.
> Thank you once again for your support and constructive feedback. We hope the response above effectively addresses your concerns.

---

> > ### Comment · Reviewer_abWa · 2025-08-04
> >
> > Thanks authors for the rebuttal, I'd keep my rating as Accept

---

### Official Review · Reviewer_88gk · 2025-07-03

**Rating:** 5
**Confidence:** 3

**Summary:**

This paper makes the first attempt to benchmark the perspective understanding capabilities of MLLMs, covering three evaluation dimensions of perception, reasoning, and robustness. The whole dataset contains 2K+ images with 5K+ QA pairs. The authors comprehensively benchmarked various open-source/proprietary MLLMs and presented several insightful findings.

**Dataset Code Accessibility:**

Yes

**Ethical Considerations:**

No, there are no or only very minor ethics concerns

**Final Justification:**

Thanks for the authors' rebuttal. I have raised the overall rating score.

**Limitations Weaknesses:**

1. The dataset is limited to multi-choice QA formats, which may not fully exploit the MLLM capabilities.
2. The annotation process is rather important, which should be introduced more detailedly. For example, what are the background and expertise of the annotators, how many annotators participate in the labeling work. More statistics about the raw labeling data should be provided.
3. The robustness metric (as formulated in Eq. 1) assumes binary consistency. Perhaps graded measures (e.g., confidence scores) could provide deeper insights.

**Strengths Contributions:**

1. This work is mostly featured by the novel benchmark design. Evaluating perspective understanding is critical for guiding the subsequent improvement of MLLMs.
2. The evaluation tasks are well-structured and enable fine-grained analyses.
3. Comprehensive coverage of various existing MLLMs and insightful findings provided.

---

> ### Author Rebuttal · Authors · 2025-07-31
>
> Dear Reviewer 88gk,
>
> Thank you for your review and valuable feedback. We are delighted that you recognized the novelty of our work, particularly the benchmark design, the well-structured evaluation tasks, and the comprehensive evaluation of MLLMs that led to insightful findings.
>
> Here is our point-by-point response to your concerns:
>
> ### **Q1: The dataset is limited to multi-choice QA formats.**
>
> We agree that the multiple-choice question answering (MCQA) format is one specific way to assess MLLM capabilities, and that open-ended generative tasks could reveal other facets of a model's abilities. Our choice of the MCQA format was a deliberate design decision based on several considerations:
>
> - **Objectivity and Scalability**: The MCQA format allows for automated, objective, and large-scale evaluation, avoiding the subjectivity and high cost associated with evaluating open-ended responses.
>
> - **Controlled Probing**: This format enables us to precisely probe a model's understanding of specific geometric concepts (e.g., distinguishing between "two-point" and "three-point" perspective) without the noise from variations in natural language generation.
>
> - **A Foundational First Step**: As the first benchmark in this domain, we believe that establishing a controlled and rigorous evaluation framework is a critical first step. It lays a solid foundation for future work that can build upon our benchmark with more complex, generative tasks.
>
> We will explicitly acknowledge this in the limitations section of our paper and frame it as a direction for future work.
>
> ### **Q2: The annotation process should be introduced with more details.**
>
> Thank you for highlighting the importance of transparency in our annotation process. We provided some details in the paper, but are happy to elaborate further.
>
> - **Process and Quality Control**: As described in Section 2.3 under "Quality Control," for subjective tasks involving spatial reasoning, we ensured that "at least two annotators independently label each sample, with disagreements resolved through discussion and consensus".
> - **Annotator Background and Expertise**: To further clarify, our annotation process involved a valuable collaboration between our internal research team and an external team of domain experts. Our internal team consists of 12 graduate students with backgrounds in computer vision, who received specific training on perspective principles using our custom annotation tool (described in Appendix D). Additionally, we partnered with a firm specializing in artistic perspective training. Their team of professional instructors, after learning about our project, generously contributed a portion of highly specialized annotations on a pro bono basis. This collaboration ensured our benchmark benefits from both technical computer vision oversight and deep, practical expertise in geometric perspective, guaranteeing a high quality of annotation.
> - **Annotation Tool**: We developed a dedicated annotation interface (shown in Figure 39) that integrates geometric drawing utilities and structured answer selection to ensure annotation accuracy and consistency.
> We will add these specific details regarding the annotator background, expertise, and training to Section 2.3 in the final manuscript to enhance the transparency and reproducibility of our data collection process.
>
> ### **Q3: The robustness metric assumes binary consistency.**
>
> We completely agree with your assessment that our binary robustness score, as formulated in Eq. 1, is indeed stringent and that a graded score would offer more nuanced insights. We appreciate your valuable suggestion.
>
> Our reason for choosing the original metric was to set a high yet principled standard for "genuine geometric consistency." The goal was to distinguish a model's intrinsic understanding of geometry from a fragile dependence on surface-level visual patterns. If a model truly understands perspective, its judgment should remain unaffected by perspective-invariant operations. Any single failure reveals a fundamental brittleness, which our binary metric was designed to detect.
>
> Nevertheless, we recognize that a graded score reveals valuable information about partial resilience. To incorporate your feedback, we have implemented a new "Graded Robustness" score and re-evaluated the models. Specifically, for each image set in the robustness task (comprising the original image and all its perturbed versions), a graded score is calculated as the percentage of variants the model answered correctly. For instance, a model that correctly answers 4 out of 5 variants now receives a graded score of 0.8 for that set, whereas its binary score would have been 0. This method directly captures the partial consistency you described.
>
> In the final revision of our paper, we will add this new "Graded Robustness" column to Table 1 to update our main results. It will be displayed next to the original binary score, offering richer details as you recommended.
>
> | Model (<7B)          | Graded P'Robust  | Binary P'Robust  | Model (7B–9B)         | Graded P'Robust  | Binary P'Robust  |
> |----------------------|------------------------|-------------------------------|------------------------|------------------------|-------------------------------|
> | InternVL2.5-2B       | 59.1                   | 46.5                          | Eagle-X4-8B            | 60.7                   | 55.3                          |
> | InternVL3-2B         | 39.0                   | 23.9                          | Qwen2-VL-7B            | 46.9                   | 25.5                          |
> | InternVL2.5-4B       | 25.0                   | 20.6                          | InternVL2.5-8B         | 38.7                   | 22.3                          |
> | Qwen2.5-VL-3B        | 22.2                   | 6.4                           | LLaVA-OV-7B            | 36.1                   | 15.9                          |
> | InternVL2-2B         | 19.3                   | 12.3                          | Qwen2.5-VL-7B          | 33.2                   | 15.3                          |
> | InternVL2.5-1B       | 19.0                   | 18.2                          | InternVL3-8B           | 29.1                   | 15.9                          |
> | InternVL2-1B         | 18.2                   | 6.7                           | InternVL2-8B           | 19.9                   | 7.9                           |
> | Qwen2-VL-2B          | 18.0                   | 4.7                           | InternVL3-9B           | 19.2                   | 7.3                           |
> | InternVL3-1B         | 16.1                   | 13.8                          | Eagle-X5-7B            | 18.4                   | 15.9                          |
> | InternVL2-4B         | 14.5                   | 7.9                           | LLaVA-Next-m-7B        | 17.9                   | 16.4                          |
> | LLaVA-OV-1B          | 13.0                   | 7.8                           | LLaVA-Next-v-7B        | 16.7                   | 16.4                          |
>
> ---
>
> | Model (10B–30B)       | Graded P'Robust  | Binary P'Robust  | Model (30B–69B)        | Graded P'Robust  | Binary P'Robust  |
> |-----------------------|------------------------|-------------------------------|-------------------------|------------------------|-------------------------------|
> | Eagle-X4-13B          | 60.7                   | 53.8                          | Qwen2.5-VL-32B          | 48.8                   | 25.5                          |
> | LLaVA-Next-13B        | 59.7                   | 51.1                          | InternVL2.5-38B         | 41.6                   | 19.1                          |
> | InternVL2.5-26B       | 52.9                   | 33.7                          | InternVL2-40B           | 29.5                   | 12.6                          |
> | InternVL2-26B         | 44.1                   | 26.5                          | InternVL3-38B           | 23.9                   | 9.1                           |
> | InternVL3-14B         | 27.3                   | 13.5                          | Eagle-X5-34B            | 18.7                   | 16.0                          |
>
> ---
>
> | Model (>70B)          | Graded P'Robust  | Binary P'Robust  | Model (Proprietary)     | Graded P'Robust  | Binary P'Robust  |
> |-----------------------|------------------------|-------------------------------|--------------------------|------------------------|-------------------------------|
> | LLaVA-OV-72B          | 71.8                   | 53.1                          | GPT-4o                   | 71.9                   | 49.9                          |
> | InternVL2.5-72B       | 56.3                   | 29.7                          | GPT-4o (CoT)             | 69.4                   | 44.4                          |
> | LLaVA-Next-72B        | 55.6                   | 33.2                          | Gemini-2-flash           | 56.8                   | 30.7                          |
> | Qwen2-VL-72B          | 51.3                   | 25.0                          | Gemini-2-flash (CoT)     | 50.5                   | 24.8                          |
> | Qwen2.5-VL-72B        | 49.9                   | 24.3                          | Gemini-1.5-flash (CoT)   | 37.8                   | 11.6                          |
> | InternVL2-72B         | 43.9                   | 22.9                          | GPT-4o-mini              | 28.7                   | 10.8                          |
> | InternVL3-78B         | 43.6                   | 25.5                          | Gemini-1.5-flash         | 20.6                   | 10.6                          |
>
> ---
>
> Thank you again for your constructive feedback. We hope the response above effectively addresses your concerns.

---

> > ### Comment · Reviewer_88gk · 2025-08-07
> >
> > The authors have effectively addressed my questions.

---

> > > ### Author Response · Authors · 2025-08-07
> > >
> > > Dear Reviewer 88gk,
> > >
> > > We greatly appreciate your thoughtful engagement and careful reading of our rebuttal! If our rebuttal has effectively addressed your concerns and clarified the points raised in your initial review, we would be grateful if you could kindly consider increasing your overall rating.
> > >
> > > Thank you again for your valuable feedback and your contribution to the review process!
> > >
> > > Best,
> > >
> > > Authors

---

### Official Review · Reviewer_zkum · 2025-07-03

**Rating:** 5
**Confidence:** 3

**Summary:**

This paper introduces MMPerspective, the first benchmark created to systematically evaluate how well Multimodal Large Language Models (MLLMs) understand visual perspective. The benchmark is structured across three dimensions: Perspective Perception, Perspective Reasoning, and Perspective Robustness, which are broken down into 10 distinct tasks. To support this evaluation, the authors curated a dataset of 2,711 images and 5,083 question-answer pairs. Through a large-scale evaluation of 43 MLLMs, the study reveals significant limitations; while models perform reasonably well on basic perceptual tasks, they falter on more complex reasoning and struggle to maintain consistent predictions under perspective-preserving image transformations. The analysis further shows that while overall accuracy tends to improve with model size, robustness does not scale as effectively, pointing to fundamental challenges in developing truly geometry-aware AI systems.

**Additional Feedback:**

No.

**Dataset Code Accessibility:**

Yes

**Dataset Code Comments:**

Partly, The author provides a link to the data, but not a link to the code

**Ethical Comments:**

I didn't identify the significant ethical concerns in the paper.

**Ethical Considerations:**

No, there are no or only very minor ethics concerns

**Final Justification:**

The paper’s primary strength lies in its novel benchmark design. Evaluating perspective understanding is crucial for advancing the development of MLLMs.

**Limitations Weaknesses:**

1. The choice of a binary, "all-or-nothing" robustness score is a significant limitation. This metric, which awards a point only if a model is correct on the original image and all its perturbations, is overly punitive and lacks nuance. For example, a model that correctly identifies the perspective type in the original image and 2 out of 3 perturbations (e.g., cropping and masking) but fails on the flipped version receives a score of 0. This is the same score given to a model that fails on every single image in the set. This approach discards valuable information about partial or graded robustness, preventing a deeper understanding of how and why models lose consistency. A more informative approach would be to report a graded score, such as the average accuracy across all variants of an image. This would reveal which models are more resilient, even if not perfectly consistent, and could highlight which specific transformations are most challenging for current architectures.
2. The paper would be significantly strengthened by a deeper, more qualitative investigation into why models fail. The current analysis focuses on quantitative error patterns (e.g., Figure 7), but it doesn't provide concrete examples of model failures that would build intuition. For instance, the Critical Line Perception (CLP) task proved difficult for all models, with even the best models performing poorly. The paper could have included an example from this task, showing the image, the candidate lines, and the incorrect line chosen by a high-performing model. This could be followed by a hypothesis for the error, such as the model being distracted by a visually salient but geometrically irrelevant line. Such examples would move the analysis from "what" models get wrong to "why," offering more direct avenues for future improvement.
3. The analysis of Chain-of-Thought (CoT) prompting, while interesting, is somewhat brief. The paper shows that CoT provides an average accuracy gain of +5.59%, which is a positive but modest improvement. The analysis could be more critical by exploring the variance in these gains. Table 2 shows that for some tasks and models, CoT provides a large benefit (e.g., +24.59% for Gemini-1.5-flash on PTR), while for others, it slightly degrades performance (e.g., -1.99% for Gemini-2-flash on LRR). A deeper dive into when and why CoT fails or succeeds in the context of spatial reasoning would be a more valuable contribution than the current, more general conclusion that it is broadly beneficial.

**Strengths Contributions:**

1. The paper tackles a fundamental, yet overlooked, aspect of visual understanding. Perspective is crucial for interpreting 3D space from 2D images. This work provides a valuable tool for the community.
2. The MMPerspective benchmark is thoughtfully designed. The taxonomy of Perception, Reasoning, and Robustness creates a logical hierarchy of difficulty and allows for a nuanced analysis of model capabilities. The 10 tasks are well-motivated and effectively probe different facets of perspective understanding, from low-level feature detection to high-level spatial inference.
3. The paper presents several key findings based on a detailed analysis of how 43 different Multimodal Large Language Models (MLLMs) performed on the MMPerspective benchmark: 1) Scaling laws show mixed results. 2) Chain-of-Thought (CoT) prompting offers benefits.  3) Error patterns reveal architectural biases and universal challenges.

---

> ### Author Rebuttal · Authors · 2025-07-31
>
> Dear Reviewer zkum,
>
> Thank you for your valuable and constructive feedback. We are pleased that you recognize the contribution of our work in addressing a fundamental yet overlooked aspect of visual understanding, and that you appreciate the thoughtful design of our benchmark, with its hierarchical taxonomy, and the value of our key findings.
>
> We would like to address your concerns point by point.
>
> ### **Q1: The limitation of the binary, "all-or-nothing" robustness score.**
>
> We completely agree with your assessment that our binary robustness score is indeed stringent and that a graded score would offer more nuanced insights. Our rationale for choosing the original metric was to establish a high but principled standard for "genuine geometric consistency." The goal was to separate a model's intrinsic understanding of geometry from a brittle reliance on surface-level visual patterns. If a model truly comprehends perspective, its judgment should not be altered by perspective-invariant operations. Any single failure indicates a fundamental brittleness, which our binary metric was designed to capture.
>
> Nevertheless, we recognize that a graded score reveals valuable information about partial resilience. To incorporate your feedback, we have implemented a new "Graded Robustness" score and re-evaluated the models. Specifically, for each image set in the robustness task (comprising the original image and all its perturbed versions), a graded score is calculated as the percentage of variants the model answered correctly. For instance, a model that correctly answers 4 out of 5 variants now receives a graded score of 0.8 for that set, whereas its binary score would have been 0. This method directly captures the partial consistency you described.
>
> In the final version of our paper, we will update our main results in Table 1 to include this new "Graded Robustness" column. It will be presented alongside the original binary/all-or-nothing score to provide the richer, more nuanced context you suggested.
>
> | Model (<7B)          | Graded P'Robust  | All-or-Nothing P'Robust  | Model (7B–9B)         | Graded P'Robust  | All-or-Nothing P'Robust  |
> |----------------------|------------------------|-------------------------------|------------------------|------------------------|-------------------------------|
> | InternVL2.5-2B       | 59.1                   | 46.5                          | Eagle-X4-8B            | 60.7                   | 55.3                          |
> | InternVL3-2B         | 39.0                   | 23.9                          | Qwen2-VL-7B            | 46.9                   | 25.5                          |
> | InternVL2.5-4B       | 25.0                   | 20.6                          | InternVL2.5-8B         | 38.7                   | 22.3                          |
> | Qwen2.5-VL-3B        | 22.2                   | 6.4                           | LLaVA-OV-7B            | 36.1                   | 15.9                          |
> | InternVL2-2B         | 19.3                   | 12.3                          | Qwen2.5-VL-7B          | 33.2                   | 15.3                          |
> | InternVL2.5-1B       | 19.0                   | 18.2                          | InternVL3-8B           | 29.1                   | 15.9                          |
> | InternVL2-1B         | 18.2                   | 6.7                           | InternVL2-8B           | 19.9                   | 7.9                           |
> | Qwen2-VL-2B          | 18.0                   | 4.7                           | InternVL3-9B           | 19.2                   | 7.3                           |
> | InternVL3-1B         | 16.1                   | 13.8                          | Eagle-X5-7B            | 18.4                   | 15.9                          |
> | InternVL2-4B         | 14.5                   | 7.9                           | LLaVA-Next-m-7B        | 17.9                   | 16.4                          |
> | LLaVA-OV-1B          | 13.0                   | 7.8                           | LLaVA-Next-v-7B        | 16.7                   | 16.4                          |
>
> ---
>
> | Model (10B–30B)       | Graded P'Robust  | All-or-Nothing P'Robust  | Model (30B–69B)        | Graded P'Robust  | All-or-Nothing P'Robust  |
> |-----------------------|------------------------|-------------------------------|-------------------------|------------------------|-------------------------------|
> | Eagle-X4-13B          | 60.7                   | 53.8                          | Qwen2.5-VL-32B          | 48.8                   | 25.5                          |
> | LLaVA-Next-13B        | 59.7                   | 51.1                          | InternVL2.5-38B         | 41.6                   | 19.1                          |
> | InternVL2.5-26B       | 52.9                   | 33.7                          | InternVL2-40B           | 29.5                   | 12.6                          |
> | InternVL2-26B         | 44.1                   | 26.5                          | InternVL3-38B           | 23.9                   | 9.1                           |
> | InternVL3-14B         | 27.3                   | 13.5                          | Eagle-X5-34B            | 18.7                   | 16.0                          |
>
> ---
>
> | Model (>70B)          | Graded P'Robust  | All-or-Nothing P'Robust  | Model (Proprietary)     | Graded P'Robust  | All-or-Nothing P'Robust  |
> |-----------------------|------------------------|-------------------------------|--------------------------|------------------------|-------------------------------|
> | LLaVA-OV-72B          | 71.8                   | 53.1                          | GPT-4o                   | 71.9                   | 49.9                          |
> | InternVL2.5-72B       | 56.3                   | 29.7                          | GPT-4o (CoT)             | 69.4                   | 44.4                          |
> | LLaVA-Next-72B        | 55.6                   | 33.2                          | Gemini-2-flash           | 56.8                   | 30.7                          |
> | Qwen2-VL-72B          | 51.3                   | 25.0                          | Gemini-2-flash (CoT)     | 50.5                   | 24.8                          |
> | Qwen2.5-VL-72B        | 49.9                   | 24.3                          | Gemini-1.5-flash (CoT)   | 37.8                   | 11.6                          |
> | InternVL2-72B         | 43.9                   | 22.9                          | GPT-4o-mini              | 28.7                   | 10.8                          |
> | InternVL3-78B         | 43.6                   | 25.5                          | Gemini-1.5-flash         | 20.6                   | 10.6                          |
>
> ---
>
> ### **Q2: Lack of a deeper, qualitative investigation into model failures.**
>
> We appreciate your suggestion to include qualitative analysis to explain why models fail, not just that they fail.
>
> We would like to gently point out that we included such an analysis of representative failure cases in Appendix C.2, covering examples from GPT-4o and Gemini-2-flash on tasks like Perspective Type Reasoning, Line Relationship, and Perspective Transformation Spotting (see Figures 33, 34, and 35). Our analysis there concluded that "the models made direct factual errors when analyzing the information in the images, rather than logical errors during the CoT process". This result directly supports your intuition that the root cause lies in limitations in core visual understanding.
>
> To make this critical finding more visible, we will move one of the most illustrative qualitative failure examples (e.g., from a complex task like CLP or PTR, as you suggested) from the appendix into the main body of the paper in the final version, to better highlight the specific failure modes of the models.
>
> ### **Q3: The analysis of CoT prompting.**
>
> You make an excellent point about the need for a more critical discussion of the variance in CoT's effectiveness. As you noted, Table 2 highlights this variability, with significant gains in some cases (e.g., +24.59% for Gemini-1.5-flash on PTR) and minor degradation in others. Our analysis of failure cases in Appendix C.2 further suggests that the model's initial visual perception caps CoT's effectiveness a model fails to correctly "see" or interpret the geometric cues in an image in the first place (i.e., makes a "factual error"), even a perfect step-by-step reasoning process cannot salvage the outcome.
>
> In the final version, we will expand our discussion on CoT in the main text. We will explicitly analyze the variance in its performance gains and link its limitations to the model's foundational weaknesses in visual perception, drawing from the evidence presented in our failure case analysis.
>
> Thank you again for your time and insightful feedback. We hope the response above addresses your concerns.

---

> > ### Comment · Reviewer_zkum · 2025-08-04
> >
> > After reading the rebuttal and considering the comments from other reviewers, I believe the authors have fully addressed the raised concerns.
> >
> > In my view, the paper’s primary strength lies in its novel benchmark design. Evaluating perspective understanding is crucial for advancing the development of MLLMs. The newly provided results and analyses effectively resolve the earlier concerns.
> >
> > Accordingly, I am increasing my score from a 4 (weak accept) to a 5 (accept) in support of acceptance.

---

### Decision · Program_Chairs · 2025-09-18

**Decision:**

Accept (poster)

**Comment:**

The paper presents a dataset of over 5K prompts spanning 2K images to understand perspective in vision language models. They benchmark 40+ models and provide an excellent analysis on limitations of current models. All reviewers are in agreement that the paper should be accepted to the conference and I concur. There were some common questions raised (feasibility of MCQ format, prompting / sampling temperature details, issues with the binary 0/1 metric), but the reviewers have addressed them adequately.